# The future of fish in Africa: Employment and investment opportunities

**Chin Yee Chan** [ID]⊕1☯*, **Nhuong Tran** [ID]⊕1☯*, **Kai Ching Cheong** [ID]1, **Timothy B. Sulser** [ID]⊕2, **Philippa J. Cohen**1, **Keith Wiebe**2, **Ahmed Mohamed Nasr-Allah**3

**1** WorldFish, Penang, Malaysia, **2** International Food Policy Research Institute, Washington, DC, United States of America, **3** WorldFish, Sharkia, Egypt

☯ These authors contributed equally to this work.
* c.chan@cgiar.org (CYC); n.tran@cgiar.org (NT)

**Data Availability Statement:** Data are available on GitHub at https://github.com/IFPRI/IMPACT.

**Funding:** NT and CC received funding from the CGIAR Research Programs on: Fish Agri-Food

## Abstract

One of the most pressing challenges facing food systems in Africa is ensuring availability of a healthy and sustainable diet to 2.4 billion people by 2050. The continent has struggled with development challenges, particularly chronic food insecurity and pervasive poverty. In Africa's food systems, fish and other aquatic foods play a multifaceted role in generating income, and providing a critical source of essential micronutrients. To date, there are no estimates of investment and potential returns for domestic fish production in Africa. To contribute to policy debates about the future of fish in Africa, we applied the International Model for Policy Analysis of Agriculture Commodities and Trade (IMPACT) to explore two Pan-African scenarios for fish sector growth: a business-as-usual (*BAU*) scenario and a high-growth scenario for capture fisheries and aquaculture with accompanying strong gross domestic product growth (*HIGH*). Post-model analysis was used to estimate employment and aquaculture investment requirements for the sector in Africa. Africa's fish sector is estimated to support 20.7 million jobs in 2030, and 21.6 million by 2050 under the *BAU*. Approximately 2.6 people will be employed indirectly along fisheries and aquaculture value chains for every person directly employed in the fish production stage. Under the *HIGH* scenario, total employment in Africa's fish food system will reach 58.0 million jobs, representing 2.4% of total projected population in Africa by 2050. Aquaculture production value is estimated to achieve US$ 3.3 billion and US$ 20.4 billion per year under the *BAU* and *HIGH* scenarios by 2050, respectively. Farm-gate investment costs for the three key inputs (fish feeds, farm labor, and fish seed) to achieve the aquaculture volumes projected by 2050 are estimated at US$ 1.8 billion per year under the *BAU* and US$ 11.6 billion per year under the *HIGH* scenario. Sustained investments are critical to sustain capture fisheries and support aquaculture growth for food system transformation towards healthier diets.

## Introduction

Ensuring that a healthy and sustainable diet is available to 2.4 billion Africans by 2050 is one of the most pressing challenges facing Africa's food systems [1–4]. The continent has struggled with a series of interconnected development challenges, particularly in fighting chronic food

Systems (FISH) led by WorldFish (https://fish.cgiar.org); Policies, Institutions, and Markets (PIM) led by International Food Policy Research Institute (IFPRI) (https://pim.cgiar.org); and Climate Change, Agriculture and Food Security (CCAFS) led by Alliance of Bioversity International and International Center for Tropical Agriculture (https://ccafs.cgiar.org). These programs are supported by contributors to the CGIAR Trust Fund. The funders had no role in study design, data collection and analysis, decision to publish, or preparation of the manuscript.

**Competing interests:** The authors have declared that no competing interests exist.

insecurity and overcoming pervasive poverty–the two foundational Sustainable Development Goals [5]. In Africa's food systems, fish and other aquatic foods play a multifaceted role as a way of life, generating income, and providing a critical source of essential micronutrients, particularly for women and infants [1, 6–9]. Nevertheless, the current and future values of fish and aquatic foods in Africa are often overlooked in development research, policy and investment. It is argued that this oversight means multiple pathways to address malnutrition and food insecurity are underexplored [10].

Fish consumed in Africa are predominantly provided by capture fisheries sourced from rivers, large inland lakes and coastal systems [11]. Whereas aquaculture is one of the fastest growing food production sectors globally [12, 13], Africa contributed only 2.7% to the global aquaculture share in 2019. Nevertheless, the African aquaculture sector is maintaining double digit average annual growth rates in the last two decades in response to increasing fish demand in the continent [12]. Despite this growth, capture fisheries and aquaculture do not supply sufficient fish and there is a significant gap between fish supplies and consumer demand in Africa [1, 8]. Further, the fish supply gap is projected to widen due to a dramatic increase in fish demand, driven by rapid population and income growth, diet transformation resulting from urbanization, and changing consumer preferences [1, 8, 14]. In addition to these growing demands, unmet nutritional needs persist and continue to increase, particularly for women of reproductive age, children under the age of five and in the first 1000 days of life [15]. Increasing fish supply, reducing waste and loss, supporting intra-regional and international fish trade, and ensuring equitable distribution and access to fish are important strategies to address some dietary deficiencies and the costly individual and societal consequences [7, 16, 17].

Africa hosts regions that are amongst the most susceptible to global climate change [18]. Climate change projections [19–22] indicate that most of northern and southern Africa will experience high water stress while eastern, central, and western Africa will be subject to increasingly heavy rains and flooding [23, 24]. Changes in precipitation and temperature patterns due to climate change will create further stress in inland lake, river and oceanic ecosystems with ramifications on fish supply and the broader wellbeing of actors in the food systems. Climate change is projected to reduce the potential fisheries catch in the Exclusive Economic Zones (EEZs) in Africa [21]. Coupling with climate change impacts, the activities of foreign fishing vessels in African EEZs are also likely to impact fish availability and access in Africa [25]. In sum, there are growing uncertainties and daunting challenges associated with the future of Africa's food systems associated with large-scale drivers that operate outside and within fisheries and aquaculture systems. These challenges, amplified by the unprecedented COVID-19 pandemic have led the governments of African and regional organizations to determine the potential investment opportunities and interventions in fisheries and aquaculture to address food and nutrition security [26, 27].

Public and private sector investment will be critical to secure diverse supplies of fish and other aquatic foods from capture fisheries and aquaculture. Whilst contributing to food and nutrition security in Africa will require four simultaneous strategies (i.e., increasing fish supply, reducing waste and loss, supporting fish trade, and ensuring equitable distribution and access), in this paper we focus on fish supply that could be achieved by increasing production. To date, there are no estimates of investment and potential returns for domestic fish production in Africa. To contribute to policy debates about the future of fish in Africa, we develop two scenarios; *business-as-usual* (*BAU*) and *high capture fisheries and aquaculture with stronger GDP growth* (*HIGH*). We first project future fish supply and demand in Africa to 2050 using the International Model for Policy Analysis of Agriculture Commodities and Trade (IMPACT). Second, we conduct post-model analysis to extrapolate future potential direct (capture fisheries and aquaculture) and indirect employment that would be associated with the

*BAU* and *HIGH* scenarios. Finally, we estimate future aquaculture production value and investment costs required to achieve *BAU* and *HIGH* scenarios.

## Materials and methods

To provide more comprehensive and consistent outlooks and prospects of fish and aquatic food systems, efforts have been made to integrate fish into foresight modeling of agriculture and livestock commodities. We apply the IMPACT fish model developed by International Food Policy Research Institute (IFPRI), which is a partial equilibrium economic model containing a system of equations for analyzing baseline and alternative scenarios for fish demand, supply, trade and prices at global, regional and country level in responding to future changes such as income, population and technological progress. [28, 29]. Previous application of the model by the World Bank in "*Fish to 2030*" report [29] used global historical data up through 2009 to develop business-as-usual (*BAU*) scenario. The projection from that model underestimated the 2010–2015 historical trend of capture fisheries and aquaculture production. To address these shortcomings, we re-calibrate the model with recent dataset and parameters of fish production, consumption, trade, population and GDP compiled from FAO, UN and IFPRI databases [4, 12, 30]. Specifically, we revisited the productivity growth assumptions of the model, using expert knowledge informed by fisheries and aquaculture specific biophysical and socio-economic factors and fish management and production targets defined by national governments [1]. The progressive improvement of IMPACT fish model used to project future Africa's fish sector is illustrated in Fig 1.

In this study, we focused on eight African nations: Egypt, Ghana, Kenya, Malawi, Nigeria, Tanzania, Uganda and Zambia. We selected these countries because they are 1) nations projected to face the largest shortfalls in fish supply relative to demands, 2) experience high rates of fish consumption, and 3) are amongst the nations experiencing relatively rapid growth in aquaculture (Table 1). These eight countries are home to 40% of Africa's total population but produce over 95% of aquaculture and 30% of capture fisheries production (by volume) in the continent in 2019. About half of fish consumed in Africa is by these eight countries, suggesting slightly higher per capita fish consumption rates than elsewhere in Africa [31]. Among these eight countries, Uganda, Tanzania, Malawi, Kenya, and Ghana are classified by Food and Agriculture Organization (FAO) as low-income food-deficit countries [32].

We consulted 76 experts from Egypt (43%), Nigeria (32%), Tanzania (15%), Zambia (4%), Ghana (2%), Kenya (2%) and South Africa (2%), during five stakeholder consultation workshops organized in Egypt, Nigeria, and Tanzania from 2017 to 2019. We sought the input of experts from government (27%), non-governmental organizations (47%), academia (13%), and private sector (13%) from different fields of expertise, covering fisheries, aquaculture, nutrition, gender, trade and economics to update and refine the model, explore alternative scenarios, validate projection results, verify the employment and investment dataset, and verify the post-model employment and investment estimation. The consensus had reached when no further comments from the stakeholders during consultation process.

### Scenario analysis

We developed two scenarios in this study. The first scenario was business-as-usual future (*BAU*) which was characterized by a set of model parameters that reflect a continuation of past trends into the future. We had determined these trends from the regional experts we had gathered in the consultation workshops. In our *BAU* scenario, we use the Shared Socioeconomic Pathway (SSP) 2 [35], which assumes economic development continues but is not uniform, environmental degradation continues, but at a slowing pace compared to historical trends,

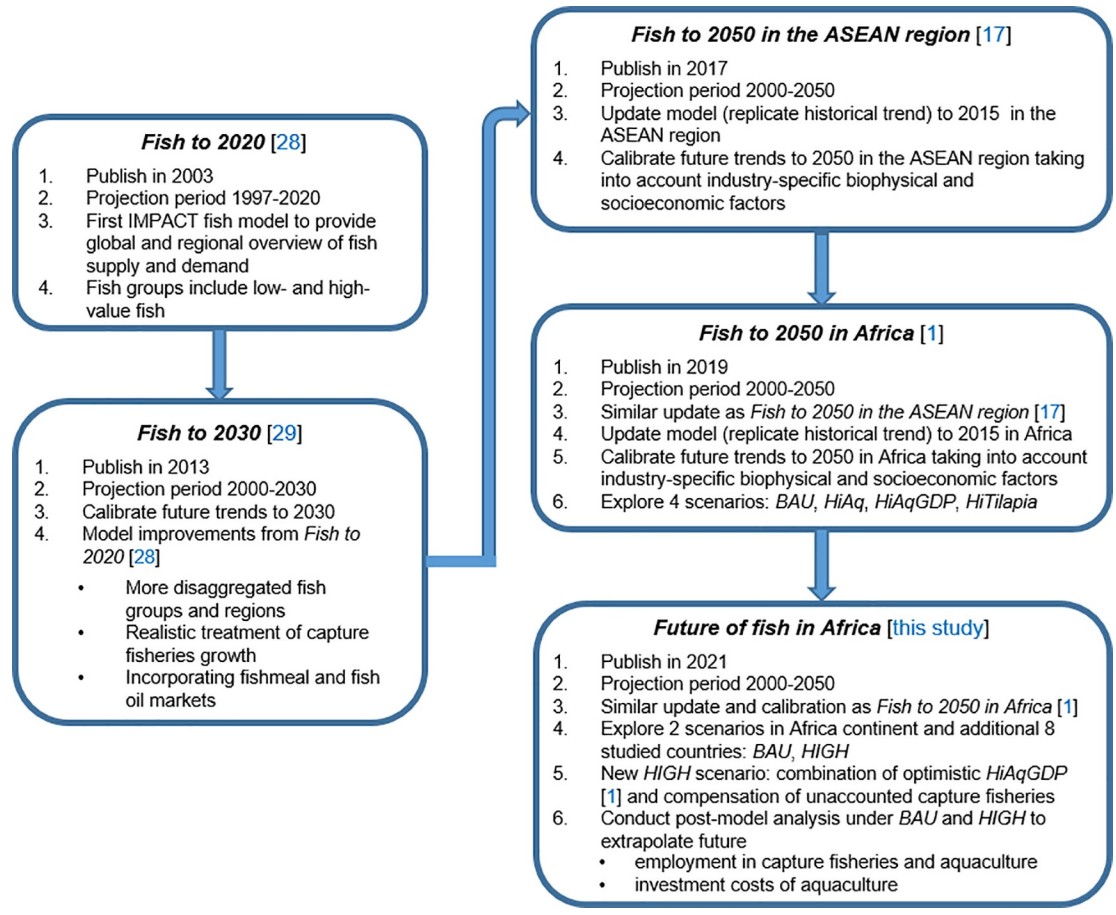

**Fig 1. Chronological model improvement and analysis using IMPACT fish model.**

and climate change presents moderate challenges to both adaptation and mitigation. Under the *BAU* scenario, African economies are assumed to have a low annual income growth rate of 2.9% from 2015–2050. This *BAU* scenario replicated projection results reported in our previous study [1].

The second alternative scenario is called *high capture fisheries and aquaculture with stronger GDP growth* (*HIGH*) assumes high aquaculture growth rates being driven by substantial investment in the industry. The model was calibrated such that aquaculture output grows at 12.7% over the 2015–2030 period (a relative improvement compared to the 10.6% aquaculture output growth observed from the 2005 to 2015 period). This is achieved by adjusting the model's exogenous productivity growth rates of the top five aquaculture producing countries in Africa (Egypt, Nigeria, Uganda, Ghana, and Zambia) for key selected species farmed in Africa (59% tilapia, 11% catfish and 11% mullet) from 2015–2050. For capture fisheries, FAO statistics [12] reported that, in 2017, Africa produced 7.0 million tonnes of marine capture fisheries and 3.0 million tonnes of inland fisheries. However, Kolding *et al.* [36] estimated substantially higher production (about 20 million tonnes) from inland fisheries based on the total freshwater resources available in Africa (e.g., lakes, rivers, reservoirs, flood plains, and swamps). Given this disparity in capture fisheries estimates, we postulate in this scenario that the potentially unaccounted capture fisheries quantities are accrued to the existing *BAU* projections. To investigate the low per capita fish consumption in Africa, this scenario also assumes an

**Table 1. Contribution of fish to food security in Africa and the world.**

| Indicator | Year | Egypt | Ghana | Kenya | Malawi | Nigeria | Tanzania | Uganda | Zambia | Studied countries | Africa | World |
|---|---|---|---|---|---|---|---|---|---|---|---|---|
| **Demographic and socio-economic status[a]** | | | | | | | | | | | | |
| Population (million) | *2020* | 102.3 | 31.1 | 53.8 | 19.1 | 206.1 | 59.7 | 45.7 | 18.4 | 536.3 | 1,340.6 | 7,794.8 |
| Population average annual growth (%) | *2010–2020* | 2.1 | 2.3 | 2.5 | 2.8 | 2.7 | 3.0 | 3.5 | 3.1 | 2.6 | 2.6 | 1.1 |
| Urban population (%) | *2020* | 42.8 | 57.3 | 28.0 | 17.4 | 52.0 | 35.2 | 25.0 | 44.6 | 42.5 | 43.3 | 56.2 |
| GDP per capita (current US$) | *2020* | 3,548 | 2,329 | 1,838 | 625 | 2,097 | 1,077 | 817 | 1,051 | 2,047 | 1,789 | 10,926 |
| GDP average annual growth (%) | *2010–2020* | 5.2 | 8.4 | 9.5 | 5.6 | 1.8 | 6.9 | 3.5 | -0.5 | 4.0 | 1.6 | 2.5 |
| Undernourishment (%) | *2019* | 5.4 | 6.1 | 24.8 | 17.3 | 14.6 | 25.1 | n.a. | n.a. | 14.6 | 17.7 | 8.9 |
| Unemployment (% total labor force) | *2020* | 10.5 | 4.5 | 3.0 | 6.0 | 9.0 | 2.2 | 2.4 | 12.2 | 7.1 | 7.7 | 6.5 |
| | Year | 2017 | *2016* | *2015* | *2016* | *2018* | *2017* | *2016* | *2015* | | | *2017* |
| Population below US$1.90 a day (%) | | 3.8 | 12.7 | 37.1 | 69.2 | 39.1 | 49.4 | 41.3 | 58.7 | n.a. | n.a. | 9.3 |
| **Contribution of fish to food supply[b]** | | | | | | | | | | | | |
| Total fish production (thousand tonnes) | *2019* | 2,039 | 445 | 144 | 163 | 1,115 | 487 | 706 | 136 | 5,235 | 12,385 | 177,834 |
| Share of aquaculture production (%) | *2019* | 80.5 | 11.8 | 12.9 | 5.1 | 26.0 | 3.4 | 14.6 | 28.3 | 41.4 | 18.4 | 48.0 |
| Aquaculture average annual growth (%) | *1999–2019* | 10.4 | 15.6 | 22.9 | 14.1 | 13.8 | 24.7 | 30.9 | 11.7 | 11.3 | 11.1 | 5.2 |
| Capture fisheries average annual growth (%) | *1999–2019* | -0.3 | -1.2 | -2.4 | 6.3 | 3.0 | 2.1 | 5.0 | 1.9 | 1.6 | 2.3 | 0.04 |
| **Contribution of fish to food and nutritional status[c]** | | | | | | | | | | | | |
| Fish consumption (kg/capita/year) | *2018* | 23.2 | 24.8 | 3.0 | 11.9 | 8.9 | 6.8 | 10.9 | 11.7 | 12.1 | 10.3 | 20.2 |
| Fish protein (g/capita/day) | *2018* | 6.6 | 8.0 | 0.9 | 3.5 | 2.6 | 2.2 | 3.3 | 3.5 | 3.6 | 3.0 | 5.6 |
| Animal protein (g/capita/day) | *2018* | 26.4 | 15.4 | 14.9 | 9.3 | 7.3 | 12.1 | 12.3 | 13.7 | 13.5 | 15.2 | 32.9 |
| Fish/animal protein (%) | *2018* | 25.0 | 52.2 | 5.7 | 37.6 | 35.3 | 18.5 | 26.5 | 25.3 | 29.1 | 20.0 | 16.9 |
| **Contribution of aquaculture value[b]** | | | | | | | | | | | | |
| Farm-gate value (million US$) | *2019* | 2,862 | 190 | 64 | 38 | 833 | 62 | 242 | 105 | 4,395 | 4,857 | 259,548 |
| Farm-gate price (US$/kg) | *2019* | 1.7 | 3.6 | 3.5 | 4.6 | 2.9 | 3.8 | 2.3 | 2.7 | 2.0 | 2.1 | 3.0 |

Author's computation from data source [a]UN [4]

[a]World Bank [33]

[b]FishStatJ [12, 31]

and

[c]FAO [34].

increase in per capita incomes. Under the *HIGH* scenario, we set a moderate optimistic annual income growth rate of 4.8% per year compared to SSP 2 of 2.9% under *BAU*.

## Post-model estimation of employment

Employment is a key indicator for assessing socio-economic contributions of the fisheries and aquaculture sectors to food, incomes, and livelihoods. Yet, due to the informal and dispersed nature of much of the sector, quality employment data are limited for both capture fisheries and aquaculture and their value chains. To estimate direct and indirect employment in the *BAU* and *HIGH* scenarios, we reviewed national employment data from global data sets [37, 38] and national sources [39–54]. We adopted the definitions of direct and indirect employment used by the FAO [38] which suggest a full time employee is one that received 90% of their livelihood or spends 90% of their time in that occupation; a part time employee between

30–90%, an occasional employee less than 30%, and indirect jobs are "those associated with ancillary activities such as the building of infrastructure (ponds, cages, tanks, etc.), feed and seed production, manufacturing of fish processing equipment, packaging, marketing, and distribution". We also take indirect employment equals direct employment (full-time equivalent number of jobs) times employment multiplier presented in Table 4.

To account for inconsistent data, we further adjusted direct and indirect employment data to better reflect our experts' assessment of labor productivity and average employment multiplier during stakeholder consultation. We compiled the labor productivity data and used it to estimate future employment through capture fisheries. Future employment in the aquaculture sector was based on the increasing trend of labor productivity in Africa's aquaculture sector observed over the past three years [37]. Assumptions for direct employment in aquaculture include a labor productivity/efficiency increase of 10% from 2018 to 2030, and again from 2030 to 2050 in Africa and the studied countries. This assumption aligns with the potential technology advancement to reduce labor requirements in aquaculture and fisheries production in the future.

### Post-model estimation of aquaculture investment costs

For investment cost extrapolation, due to data limitations in capture fisheries, we focused explicitly on aquaculture alone to determine the size of investment needed to meet the *BAU* and *HIGH* projections of production. Aquaculture production values for the base year 2016 (except Uganda and Zambia base year in 2014) were computed using commodity prices collected from literature for each country. Production values of the base and future projections in 2030, and 2050 under the *BAU* and *HIGH* scenarios were converted to 2010 constant US dollar using the World Bank's consumer price index. The investment needed to support projected value was built on key variable input costs such as seed (i.e., the broodstock, hatchlings, or fry that are spawned or caught from the wild), feed (i.e., a combination of ingredients made into a single feed for growing fish), and labor (i.e., fish farmers with full time equivalent number of direct jobs). The magnitude of costs for each scenario was ascertained through literature review and validated through our expert consultation workshops. The costs of inputs were determined using farm-gate prices, average productivity, average market size, survival rate, stocking density, feed conversion ratio, average wages, input prices, and profit margins (Table 2). We present the variation of these inputs information in single value in Table 2 after validation via the stakeholder consultation process. Future input costs were converted to constant US$ in 2010 using the consumer price index (i.e., dollar values are divided by the consumer price index of that year, and then multiplied by the index of 2010). Fixed costs such as infrastructure investment costs, and public spending for aquaculture research, development and extension, were not included in our estimation due to data limitations.

## Results

### Scenarios

Previous projections of *BAU* scenario [1] had suggested that African capture fisheries and aquaculture production will grow at 0.2% and 1.3%, respectively, from 2015 to 2050. Despite the higher growth rate of aquaculture, capture fisheries in Africa will continue to be the main contributor to total fish production until 2050, though Egypt is a notable exception. Driven by high population growth and low GDP growth, per capita fish consumption in Africa is projected to gradually drop from 10.0 kg/year in 2015 to 7.7 kg/year in 2050 under this scenario (Table 3).

**Table 2. Key parameters used for estimating the quantity and cost of key inputs in studied countries.**

| Base year | Egypt | Ghana | Kenya | Malawi | Nigeria | Tanzania | Uganda | Zambia |
|---|---|---|---|---|---|---|---|---|
| | 2016 | 2016 | 2016 | 2016 | 2016 | 2016 | 2014 | 2014 |
| Farm-gate price (US$/kg) | 0.95 | 1.45 | 2.12 | 0.98 | 1.41 | 1.96 | 1.99 | 1.78 |
| Feed conversion ratio (FCR) | 1.12 | 2 | 1.5 | 1.8 | 1.3 | 1.5 | 1.5 | 1.7 |
| Productivity (tonne/ha) | 10.8 | 2.9 | 5 | 1.8 | 4 | 10 | 10 | 1.1 |
| Average market size (g/fish) | 300 | 400 | 400 | - | 800 | - | - | - |
| Survival rate (%) | 90 | 80 | 80 | - | 80 | - | 70 | 90 |
| Stocking density (1000 pieces/ha) | 40 | - | 15.6 | 6 | 6.3 | 30 | 2.5 | 2.8 |
| Seed price (US$/1000 pieces) | 5.6 | 39 | 62.6 | 6.6 | 57 | 90 | 70 | 43 |
| Feed price (US$/kg) | 0.38 | 0.48 | 0.63 | 0.33 | 0.57 | 0.81 | 0.56 | 0.30 |
| Average wage (US$/year) | 713 | 145 | 188 | 33 | 509 | 175 | 211 | 339 |
| Profit margin (%) | 34 | 15 | 23 | 19 | 28 | 11 | 37 | 23 |
| References | [51, 55] | [39, 56–64] | [44, 52, 65–73] | [40, 41, 54, 74–76] | [70, 77–81] | [45, 48, 49, 82–86] | [43, 53, 87] | [46, 50, 88–90] |

All values are converted to constant US$ in 2010 based on World Bank's consumer price index.

Under the *HIGH* scenario, the total capture fisheries production is projected to be 76% and 74% higher in 2030 and 2050 compared to the *BAU* scenario. This is driven by better accounting for inland capture fisheries production, rather than substantial increases in capture fisheries production. The total aquaculture production is projected to be 350% and 558% higher in 2030 and 2050, respectively, compared to the *BAU* projections (Table 3). With these high growth assumptions, the aquaculture production in Africa will likely surpass capture fisheries production by 2050. High GDP growth will enable purchasing power to increase per capita fish consumption from 10 kg in 2015 to 12 kg in 2030 and 14 kg in 2050 (Table 3).

## Employment in fish sectors

Table 4 depicts that, overall, African capture fisheries and aquaculture sectors are estimated to sustain 20.7 million jobs (direct and indirect employment) in 2030, and generate 21.6 million jobs by 2050 under the *BAU* scenario. Direct employment in Africa's fish sector is estimated to remain relatively constant and only grow from 5.6 million in 2030 to 5.8 million in 2050 in the *BAU* scenario. In contrast, under the *HIGH* scenario, direct employment in capture fisheries and aquaculture will be more than double in comparison to the BAU, reaching 12.2 million by 2050, where for every person directly employed in the sector, 2.6 people will be indirectly employed. By 2050, capture fisheries and aquaculture sectors will sustain 58.0 million jobs. Under the *BAU* and *HIGH* scenarios, the total direct and indirect employment for the fish sector will represent 0.9% and 2.4%, respectively, of the total projected 2.4 billion African in 2050 [4].

**Table 3. IMPACT fish model scenario projection of fish production and per capita fish consumption for Africa in 2015, 2030, and 2050.**

| Region | Scenarios | Capture fisheries (million tonnes) | | | Aquaculture (million tonnes) | | | Per capita fish consumption (kg/person/year) | | |
|---|---|---|---|---|---|---|---|---|---|---|
| | | 2015 | 2030 | 2050 | 2015 | 2030 | 2050 | 2015 | 2030 | 2050 |
| **Africa** | *BAU* | 8.7 | 9.0 | 9.2 | 1.8 | 2.4 | 2.9 | 10.0 | 8.5 | 7.7 |
| | *HIGH* | | 15.8 | 16.0 | | 11.0 | 18.8 | | 12.1 | 14.0 |

**Table 4. Estimated direct and indirect employment of Africa's fish food system for *BAU* and *HIGH* scenarios in 2030 and 2050.**

| Country | Scenarios | Fish production (thousand tonnes) | | Direct labor productivity (tonnes/worker) | | Direct employment (thousand) | | Average employment multiplier | | Indirect employment (thousand) | | Total direct and indirect employment (thousand) | |
|---|---|---|---|---|---|---|---|---|---|---|---|---|---|
| | | 2030 | 2050 | 2030 | 2050 | 2030 | 2050 | 2030 | 2050 | 2030 | 2050 | 2030 | 2050 |
| **Africa** | *BAU* | 11,439 | 12,064 | 2.0 | 2.1 | 5,630 | 5,774 | 2.6 | 2.7 | 15,035 | 15,855 | 20,665 | 21,629 |
| | *HIGH* | 26,784 | 34,816 | 2.4 | 2.8 | 11,049 | 12,230 | 3.2 | 3.7 | 35,202 | 45,758 | 46,251 | 57,988 |
| **Egypt** | *BAU* | 1,924 | 2,169 | 12.4 | 13.4 | 156 | 161 | 1.8 | 2.0 | 288 | 324 | 443 | 485 |
| | *HIGH* | 4,977 | 7,632 | 12.8 | 14.1 | 389 | 540 | 1.7 | 2.0 | 680 | 1,078 | 1,069 | 1,617 |
| **Ghana** | *BAU* | 369 | 389 | 1.4 | 1.4 | 271 | 276 | 2.1 | 2.2 | 565 | 597 | 835 | 872 |
| | *HIGH* | 1,147 | 1,722 | 1.7 | 2.2 | 660 | 790 | 2.7 | 3.3 | 1,757 | 2,639 | 2,417 | 3,429 |
| **Kenya** | *BAU* | 209 | 213 | 1.9 | 1.9 | 111 | 113 | 2.0 | 2.0 | 217 | 222 | 328 | 335 |
| | *HIGH* | 618 | 630 | 2.7 | 2.6 | 228 | 242 | 2.8 | 2.7 | 644 | 656 | 872 | 897 |
| **Malawi** | *BAU* | 133 | 136 | 1.0 | 1.0 | 135 | 137 | 4.0 | 4.0 | 534 | 545 | 669 | 682 |
| | *HIGH* | 397 | 404 | 1.0 | 1.0 | 400 | 406 | 4.0 | 4.0 | 1,595 | 1,622 | 1,995 | 2,028 |
| **Nigeria** | *BAU* | 1,441 | 1,638 | 1.3 | 1.3 | 1,137 | 1,266 | 0.9 | 0.9 | 1,005 | 1,142 | 2,142 | 2,409 |
| | *HIGH* | 2,072 | 2,396 | 1.2 | 1.3 | 1,749 | 1,903 | 0.8 | 0.9 | 1,445 | 1,670 | 3,194 | 3,573 |
| **Tanzania** | *BAU* | 341 | 341 | 1.8 | 1.8 | 192 | 192 | 1.3 | 1.3 | 247 | 247 | 439 | 439 |
| | *HIGH* | 1,087 | 1,088 | 1.8 | 1.8 | 602 | 604 | 1.3 | 1.3 | 786 | 787 | 1,409 | 1,416 |
| **Uganda** | *BAU* | 639 | 669 | 3.6 | 3.6 | 179 | 183 | 3.9 | 4.0 | 693 | 726 | 872 | 909 |
| | *HIGH* | 2,070 | 2,151 | 3.4 | 3.4 | 616 | 632 | 3.6 | 3.7 | 2,245 | 2,333 | 2,861 | 2,965 |
| **Zambia** | *BAU* | 115 | 124 | 1.3 | 1.3 | 92 | 96 | 0.5 | 0.5 | 49 | 52 | 141 | 148 |
| | *HIGH* | 418 | 474 | 1.2 | 1.3 | 347 | 377 | 0.5 | 0.5 | 177 | 200 | 524 | 578 |

For direct employment in African aquaculture, even with increasing average labor productivity from 5.8 tonnes/worker in 2030 to 6.3 tonnes/worker in 2050, fish farmers are projected to increase to 0.3 million under *BAU* and sharply increase to 1.1 million under *HIGH* by 2050 due to the 71% increase in aquaculture production. Among the eight studied countries, Egypt has the highest labor productivity of 13–15 tonnes/worker, followed by Uganda, Nigeria, Ghana, and Zambia. Conversely, Malawi, Tanzania, and Kenya have relatively lower labor efficiency with less than one tonne/worker (Table 5).

Employment generated by capture fisheries contributes to over 90% of the total jobs in the African fish sector under *BAU*. This is mainly due to higher capture fisheries output but lower labor productivity as compared to aquaculture. Among the studied countries, Egypt again has the highest labor efficiency in capture fisheries of 9.3 tonnes/worker (Tables 5 and 6). Egypt is the only country that has a higher proportion of jobs generated by aquaculture than capture fisheries in the *HIGH* scenario (Fig 2). Nigeria is among the studied countries that generates the highest total employment in the fish sector, particularly in the capture fisheries sector. Only two countries—Nigeria and Zambia—have average employment multipliers less than one, resulting in the proportion of indirect employment being less than half of the total employment in the fish sector (Fig 3).

## Aquaculture production values and investment costs

Under the *BAU* scenario, Africa's aquaculture production is projected to reach 2.4 million tonnes, valued at US$ 2.8 billion, and 2.9 million tonnes, valued at US$ 3.3 billion in 2030 and 2050, respectively. Farm-gate investment costs for three key variable inputs of feed, labor, and fish seed to realize aquaculture production in 2030 and 2050 are shown in Table 7. The

**Table 5. Estimated direct employment of Africa's aquaculture sector for *BAU* and *HIGH* scenarios in 2030 and 2050.**

| Country | Scenarios | Aquaculture production (thousand tonnes) | | Labor productivity (tonnes/worker) | Labor productivity (tonnes/worker) | Direct employment | |
|---|---|---|---|---|---|---|---|
| | | 2030 | 2050 | 2030 | 2050 | 2030 | 2050 |
| **Africa** | *BAU* | 2,439 | 2,864 | 5.8 | 6.3 | 424,368 | 452,866 |
| | *HIGH* | 10,984 | 18,816 | | | 1,910,711 | 2,975,598 |
| **Egypt** | *BAU* | 1,594 | 1,843 | 13.3 | 14.6 | 120,303 | 126,449 |
| | *HIGH* | 4,550 | 7,210 | | | 343,318 | 494,569 |
| **Ghana** | *BAU* | 85 | 102 | 3.9 | 4.2 | 21,929 | 24,067 |
| | *HIGH* | 558 | 1,122 | | | 144,856 | 264,663 |
| **Kenya** | *BAU* | 33 | 38 | 0.6 | 0.6 | 60,080 | 62,722 |
| | *HIGH* | 33 | 45 | | | 60,063 | 73,717 |
| **Malawi** | *BAU* | 8 | 9 | 0.8 | 0.9 | 9,328 | 10,292 |
| | *HIGH* | 8 | 11 | | | 9,328 | 12,124 |
| **Nigeria** | *BAU* | 445 | 526 | 4.5 | 5.0 | 98,846 | 106,190 |
| | *HIGH* | 500 | 707 | | | 111,188 | 142,952 |
| **Tanzania** | *BAU* | 7 | 8 | 0.8 | 0.9 | 9,010 | 9,315 |
| | *HIGH* | 7 | 10 | | | 9,005 | 10,822 |
| **Uganda** | *BAU* | 134 | 154 | 5.4 | 5.9 | 25,071 | 26,207 |
| | *HIGH* | 136 | 184 | | | 25,468 | 31,305 |
| **Zambia** | *BAU* | 29 | 34 | 1.8 | 1.9 | 16,379 | 17,610 |
| | *HIGH* | 61 | 103 | | | 34,658 | 53,325 |

**Table 6. Estimated direct employment of Africa's capture fisheries sector for *BAU* and *HIGH* scenarios in 2030 and 2050.**

| Country | Scenarios | Capture fisheries production (thousand tonnes) | | Labor productivity (tonnes/worker) | Direct employment | |
|---|---|---|---|---|---|---|
| | | 2030 | 2050 | | 2030 | 2050 |
| **Africa** | *BAU* | 9,000 | 9,200 | 1.7 | 5,205,551 | 5,321,230 |
| | *HIGH* | 15,800 | 16,000 | | 9,138,634 | 9,254,313 |
| **Egypt** | *BAU* | 330 | 326 | 9.3 | 35,328 | 34,869 |
| | *HIGH* | 427 | 422 | | 45,690 | 45,231 |
| **Ghana** | *BAU* | 284 | 287 | 1.1 | 248,730 | 251,645 |
| | *HIGH* | 588 | 600 | | 515,164 | 525,665 |
| **Kenya** | *BAU* | 176 | 175 | 3.5 | 50,456 | 50,415 |
| | *HIGH* | 585 | 585 | | 168,211 | 168,170 |
| **Malawi** | *BAU* | 125 | 126 | 1.0 | 125,534 | 126,717 |
| | *HIGH* | 389 | 393 | | 385,213 | 393,786 |
| **Nigeria** | *BAU* | 996 | 1,113 | 1.0 | 1,038,293 | 1,159,992 |
| | *HIGH* | 1,572 | 1,688 | | 1,638,155 | 1,759,854 |
| **Tanzania** | *BAU* | 334 | 333 | 1.8 | 183,254 | 182,948 |
| | *HIGH* | 1,079 | 1,079 | | 592,958 | 592,680 |
| **Uganda** | *BAU* | 505 | 515 | 3.3 | 154,162 | 157,281 |
| | *HIGH* | 1,934 | 1,967 | | 590,683 | 600,731 |
| **Zambia** | *BAU* | 86 | 90 | 1.1 | 75,607 | 78,438 |
| | *HIGH* | 357 | 370 | | 312,323 | 324,017 |

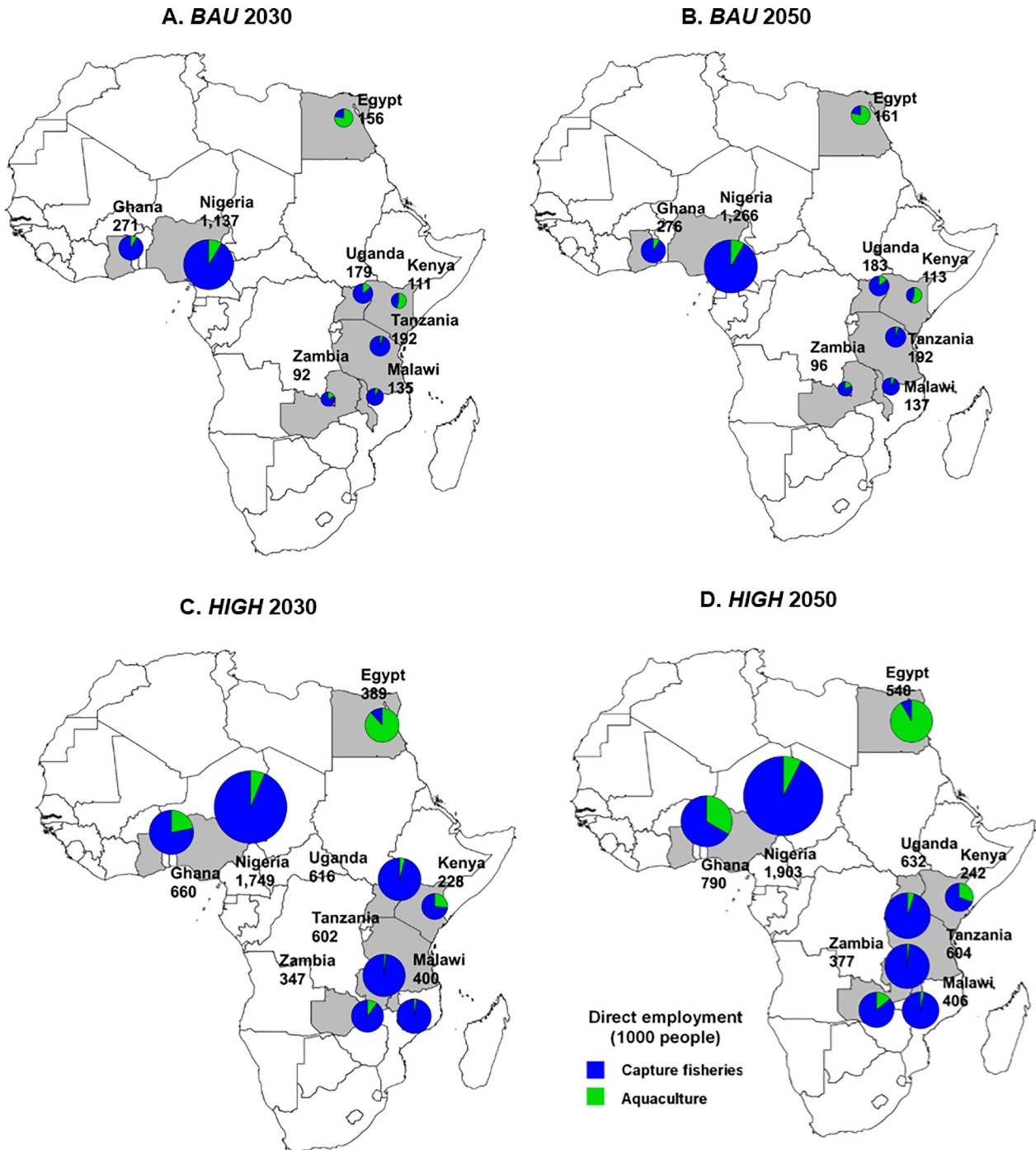

**Fig 2.** Direct employment of capture fisheries and aquaculture under *BAU* scenario in 2030 (A), *BAU* scenario in 2050 (B), *HIGH* scenario in 2030 (C), and *HIGH* scenario in 2050 (D).

investment costs are projected to increase to US$ 1.6 billion and US$ 1.8 billion in 2030 and 2050, respectively, to achieve the projected aquaculture outputs in those years. Of the three key variable costs estimated, feed costs account for 81% to 84%, labor costs range from 10% to

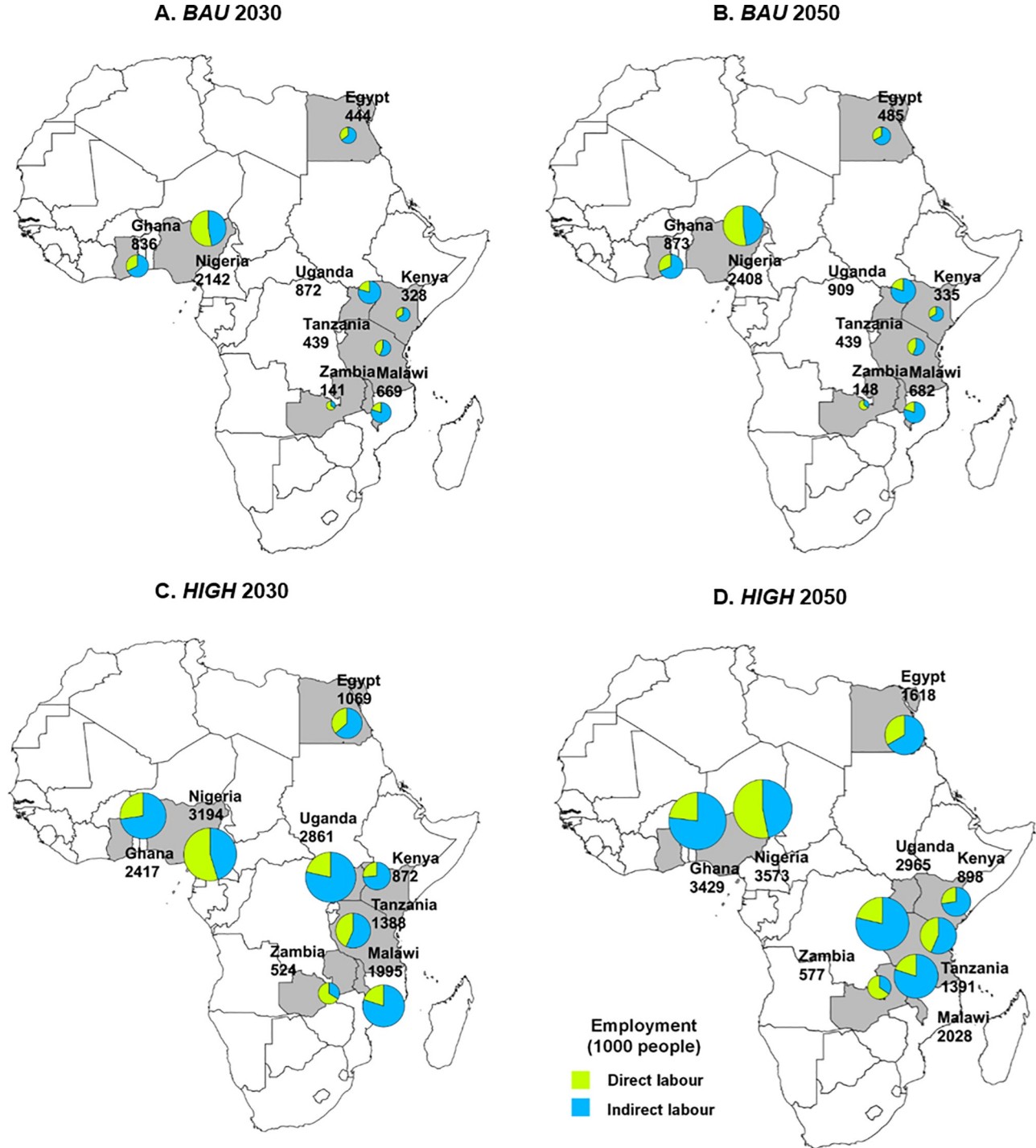

**Fig 3.** Direct and indirect employment of fish sector under *BAU* scenario in 2030 (A), *BAU* scenario in 2050 (B), *HIGH* scenario in 2030 (C), and *HIGH* scenario in 2050 (D).

12%, and fish seed costs a little over 6% (Fig 4). The investment cost structure is likely to remain the same, unless there are technological innovations in the fish feed and seed sectors, resulting in a substantial decrease in feed costs.

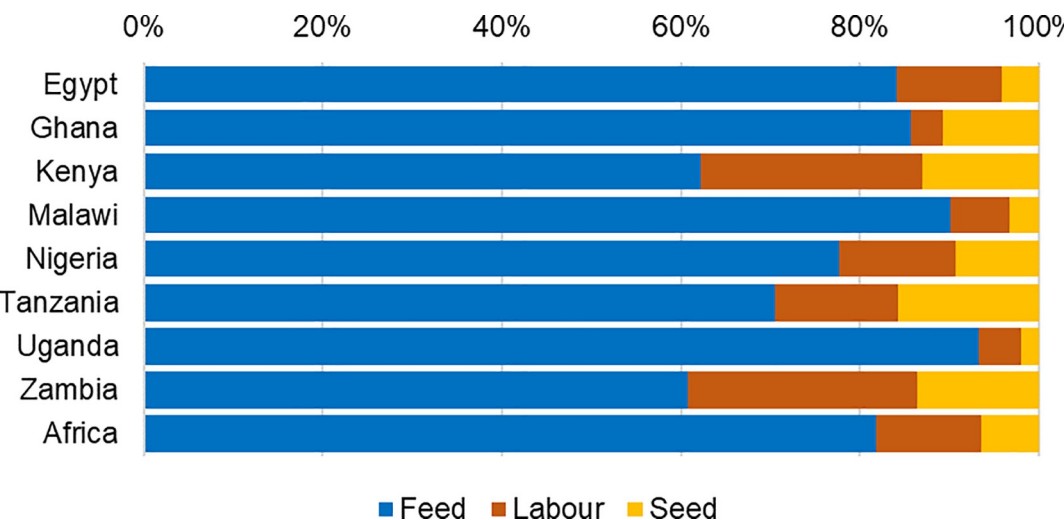

**Fig 4. Cost structure of key input production costs of aquaculture in Africa and studied countries.**

Under the *HIGH* scenarios, aquaculture production values in Africa are projected to reach US$ 11.9 billion in 2030 and US$ 20.4 billion in 2050 (Table 7). To maintain the aquaculture growth rate as projected in the *HIGH* scenario, investment costs in the three key variable costs of feed, labor, and seed need to increase to US$ 6.8 billion by 2030 and US$ 11.6 billion by 2050. These key investment costs will be invested by producers, including private aquaculture enterprises and farmers at different production scales. Similar to the *BAU* scenario, feed cost is the main component, accounting for more than 80% (Fig 4). Investing in aquaculture feed is critical to achieving the aquaculture production outputs by 2030 and 2050 projected in the *HIGH* scenario.

Our post-model estimation (Table 7) suggests that uneven distribution of future aquaculture production values and required investment costs will remain under both the *BAU* and *HIGH* scenarios. Under the *BAU*, the eight countries included in this study are projected to account for 96% of Africa's aquaculture production values in 2030 and only slightly reduce to 95% by 2050. A similar pattern is observed for investment costs of the key variable inputs required to achieve aquaculture projection output levels. The top four African aquaculture producers–namely Egypt, Nigeria, Uganda, and Ghana–account for 90% of the production values while the group of Kenya, Zambia, Tanzania, and Malawi was projected to account for 5% throughout 2030 and 2050.

## Discussion

One of Africa's biggest development challenges is to meet the nutrition needs, within sustainable limits of 2.4 billion women, men and children by 2050 [91]. Experiences in Asia and other regions show that aquatic foods, capture fisheries, and aquaculture systems [92] offer important nutritional and sustainability values, in some cases outperforming nutritional qualities of dietary supplements [93] and a relatively lower environmental footprint than animal-source foods [94, 95]. However, the role of fisheries, aquaculture and aquatic foods in the transformation of food systems has remained relatively overlooked due to the lack of scientific data, metrics, and evidence to inform donors, governments, and private investors in decision making and investment planning [96]. Using a rigorous partial equilibrium economic modeling tool, the IMPACT fish model, we generate future fish supply-demand projections in Africa to 2050

Table 7. Annual output value and key inputs costs of Africa's aquaculture for *BAU* and *HIGH* scenarios in 2030 and 2050.

| Country | Scenarios | Aquaculture production values (million US$) | | 2030 farm-gate costs (million US$) | | | | 2050 farm-gate costs (million US$) | | | |
|---|---|---|---|---|---|---|---|---|---|---|---|
| | | 2030 | 2050 | Feed | Labor | Seed | Total | Feed | Labor | Seed | Total |
| Africa | BAU | 2,799.4 | 3,290.0 | 1,313.6 | 170.6 | 101.5 | **1,585.7** | 1,545.9 | 182.4 | 120.1 | **1,848.4** |
| | HIGH | 11,862.8 | 20,373.9 | 5,670.3 | 661.7 | 421.5 | **6,753.5** | 9,859.1 | 1,010.8 | 751.7 | **11,621.6** |
| Egypt | BAU | 1,518.8 | 1,756.1 | 672.9 | 85.8 | 33.2 | **791.9** | 778.0 | 90.2 | 38.4 | **906.6** |
| | HIGH | 4,334.4 | 6,868.3 | 1,920.2 | 244.8 | 94.8 | **2,259.8** | 3,042.8 | 352.6 | 150.3 | **3,545.7** |
| Ghana | BAU | 122.7 | 148.1 | 81.8 | 3.2 | 10.2 | **95.2** | 98.7 | 3.5 | 12.3 | **114.5** |
| | HIGH | 810.2 | 1,628.4 | 540.1 | 21.0 | 67.5 | **628.6** | 1,085.6 | 38.4 | 135.7 | **1,259.7** |
| Kenya | BAU | 70.0 | 80.4 | 31.0 | 11.3 | 6.5 | **48.8** | 35.7 | 11.8 | 7.4 | **54.9** |
| | HIGH | 70.0 | 94.5 | 31.0 | 11.3 | 6.5 | **48.8** | 41.9 | 13.9 | 8.7 | **64.5** |
| Malawi | BAU | 7.7 | 9.3 | 4.6 | 0.3 | 0.2 | **5.1** | 5.6 | 0.3 | 0.2 | **6.1** |
| | HIGH | 7.7 | 11.0 | 4.6 | 0.3 | 0.2 | **5.1** | 6.6 | 0.4 | 0.2 | **7.2** |
| Nigeria | BAU | 627.6 | 741.7 | 330.2 | 50.3 | 39.5 | **420.0** | 390.2 | 54.0 | 46.6 | **490.8** |
| | HIGH | 706.0 | 998.4 | 371.4 | 56.6 | 44.4 | **472.4** | 525.3 | 72.7 | 62.8 | **660.8** |
| Tanzania | BAU | 14.5 | 16.5 | 8.9 | 1.6 | 2.0 | **12.5** | 10.2 | 1.6 | 2.3 | **14.1** |
| | HIGH | 14.5 | 19.2 | 8.9 | 1.6 | 2.0 | **12.5** | 11.8 | 1.9 | 2.6 | **16.3** |
| Uganda | BAU | 266.7 | 306.6 | 113.4 | 5.3 | 2.4 | **121.1** | 130.4 | 5.5 | 2.7 | **138.6** |
| | HIGH | 270.9 | 366.3 | 115.2 | 5.4 | 2.4 | **123.0** | 155.7 | 6.6 | 3.2 | **165.5** |
| Zambia | BAU | 51.5 | 60.9 | 14.5 | 5.6 | 3.2 | **23.3** | 17.1 | 6.0 | 3.8 | **26.9** |
| | HIGH | 109.0 | 184.3 | 30.7 | 11.8 | 6.9 | **49.4** | 51.9 | 18.1 | 11.6 | **81.6** |

All value costs are in millions of constant 2010 US$.

under the *BAU* and *HIGH* scenarios. Fish supply projections are then used to extrapolate future direct and indirect employment and investment costs needed to achieve the projected output levels.

There is a global concern that current food systems are ill-equipped to deliver nutritious food, a challenge that will be exacerbated as demands of burgeoning populations and wealth will outpace supplies. In practice, limited supplies of quality food will affect different people to different degrees based on their economic status, geography, and gender, with women of reproductive age and children under the age of five being most vulnerable to nutrient deficiencies [97]. Our *BAU* scenario projection shows that per capita fish consumption in Africa will gradually drop from 10.0 kg/person/year (about half the global and Asian fish intake) in 2015 to 7.7 kg/person/year in 2050. The drop in fish consumption we see in our *BAU* scenario is a result of population growth outpacing growth in the fish sector. Decreasing per capita fish consumption is also the outcome of modest GDP growth. Lower economic growth will constrain governments' and private sector's ability to invest in supply infrastructure, technology, and management systems that might otherwise boost supplies. In the optimistic *HIGH* scenario, with GDP growth at 4.8% per year to 2050, per capita fish consumption is projected to increase from 10.0 kg/year in 2015 to 14.0 kg/year in 2050. Investments in sustainable fisheries management and aquaculture will boost total domestic production to 34.8 million tonnes in 2050, of which the share of capture fisheries in total fisheries production in Africa will decline from 82.7% in 2015 to 46.0% in 2050. These results show that there is potential to sustain capture fisheries and expand aquaculture to meet the growing demand for fish in Africa. This needs a sound enabling macro-environment, particularly moderate to high economic growth to stimulate fish demand increase and sustained investment from farmers, investors and governments

to transform Africa's capture fisheries and aquaculture into sustainable, productive, nutrition-sensitive and inclusive aquatic food systems.

Youth employment, and the future employment of current youth, is a growing opportunity and concern globally, and aquaculture and fisheries offer possible but evolving opportunities. With rapid population growth and a young population (60% of the African population below the age of 25), it is expected that 11 million young people will enter the job market in Sub-Saharan Africa every year, while only about 3 million new jobs are created annually on the continent [98–100]. Both capture fisheries and aquaculture are important sources of employment in Africa, particularly for smallholders and value chain actors in rural areas [100]. Creating jobs in rural areas at a large-scale is critical to address these unemployment issues and income generation in Africa. Our study results show that under the *BAU* scenario, with slow aquaculture growth and almost stagnant capture fisheries, Africa's fish sector is projected to provide 22 million direct and indirect jobs by 2050. However, with the *HIGH* scenario, 58 million people will be directly and indirectly employed in fisheries and aquaculture sectors, representing 2.4% of the total projected population in Africa in 2050. The projection results indicate that growth in Africa's fish sector will create considerable employment and has the potential to generate significant income growth and facilitate inclusive value chain development to address development barriers faced by Africa. About 60 million people (14% of whom are women) were engaged in the primary sector to produce 179 million tonnes of fish globally in 2018 [101], implying a global average labor productivity of 3.0 tonnes/worker. Our projection estimated that the overall African labor productivity of direct employment in both capture fisheries and aquaculture is 2.0 tonnes/worker, slightly lower than the world average. Furthermore, employment in the fish sector in Africa will continue to be dominated by small-scale fisheries, with lower labor productivity compared to aquaculture (1.7 tonnes/worker vs. 6.3 tonnes/worker). This result highlights the importance of sustainable capture fisheries management to generate employment opportunities and provide income for the portion of the African population depending on artisanal fishing. In order to achieve the desired sustainability transformation, public policy leadership and private sector technological innovation will be required [102].

Our projection results show that strong aquaculture growth has a high potential to generate income and jobs for rural communities in Africa. Under the *HIGH* scenario, aquaculture production in Africa is projected to reach 18.8 million tonnes, generating a revenue of US$ 20.4 billion in 2050. Projected farm-gate investment costs of three key aquaculture inputs (feed, labor, and fish seed) will reach US$ 11.6 billion in 2050. It is essential to highlight that these investments can be mobilized from farmers, private sector investors and enterprises, suggesting dynamic opportunities for market-led aquaculture business development. Given that feed accounts for a major share of aquaculture production cost, this suggests that there will be bright prospects for investing in the aquaculture feed industry in Africa. It will be essential to have more supportive policies and regulations to serve as an entry point for the private sector on more inclusive ways to engage smallholders in the fish value chains.

This study provides useful insights on how aquatic foods, fisheries and aquaculture systems in Africa might evolve into the future under complex and dynamic interactions of structural changes, technological progress, income growth, and urbanization in a climate crisis. The study findings also allow drawing policy implications of different impact pathways, drivers and interventions to enhance aquatic food systems' contributions to sustainable development goals in Africa. As documented in a previous report [103], these results could be the practical usage by a wide range of stakeholders from international organizations, academic and national government. Notwithstanding these contributions, our study has several limitations due to data gaps. First, in aquaculture investment cost extrapolation, we do not estimate the required

investments for farm or value chain infrastructure. Second, we are unable to project investment costs needed for capture fisheries monitoring, management and capacity building, which mostly come from public funding and development and conservation funding. Future follow-up studies should investigate aquaculture infrastructure cost requirements and investment costs for capture fisheries management in Africa. Third, our extrapolation of outcomes focuses only on employment opportunities. Further research is needed to extend the post-model analysis to examine implications on other outcome areas such as gender equity, nutrition and environmental sustainability associated with different future projected trends. Effective and efficient use of data collection tools for gender and youth assessment needs to be embedded in a future inclusive development process. Fourth, our estimation of investment costs does not include public investment in infrastructure, human capital and research capacity needed to create macro and micro enabling environment for aquaculture and fisheries sector performance. Finally, aquatic foods are relatively new in the realm of foresight modeling tools compared to crops and livestock, and further advancement of fish foresight modeling tools is essential to improve the quality of modeling projections and incorporating these outcomes in future analyses. Given the high diversity in wild-caught and cultured fish species in Africa and worldwide, the current IMPACT model is highly aggregated with sixteen fish categories on the supply side and nine categories on the demand side. The model is calibrated using data in 2000 as a base year. This is quite out-of-date given that fisheries and aquaculture are complex and very dynamic, experiencing rapid growth over the last two decades. The IMPACT fish model uses generic assumptions to obtain parameters for specifying the fish sector equations, whereas fish and aquatic food systems are highly heterogeneous and complex. There are numerous fish types, classification schemes, and production methods. It is necessary to conduct disaggregated modeling studies for specific fish types to capture the diversity of trends within specific sub-sectors. Follow-up foresight modeling analysis and projection could address these disaggregation and complexity issues.

## Conclusions

Our current food systems face severe challenges in achieving equitable access to healthy, nutritious food, maintaining environmental sustainability, and building resilience to shocks. Fish and aquatic foods offer significant potential in the transformation of food systems toward healthy and sustainable diets, sustaining livelihoods, and generating income. The fish sectors are important for employment creation in Africa, yet, their role has been overlooked, resulting in insufficient investment to support the sector growth and sustainable system transformation to meet the increasing demand for fish. This study provides insights into future fish supply and demand projections in Africa under the *BAU* and *HIGH* scenarios and provides first estimates of employment generated and the necessary input cost investments required to secure projected fish supplies in 2030 and 2050. The study suffers limitations that should be addressed in the future. Nonetheless, this is a key and first preliminary analysis to look at macro-level employment and investment scenarios of fish sectors in Africa.

## Acknowledgments

The authors would like to thank the contribution of a wide range of African government, academia, non-profit organizations who provide valuable information and feedback to this research work. In particular, Dr. Richard Abila from IFAD; Dr. Alexander Shula Kefi from Department of Fisheries Zambia; Dr. Diaa Al-Kenawy from WorldFish Egypt office, Dr. Bernadette Tosan Fregene from University of Ibadan, Nigeria; Dr. Emmanuel Nii Abbey from University of Ghana, Dr. Mafaniso Hara from University of Western Cape, Dr. Amon Paul

Shoko from Tanzania Fisheries Research Institute, and Dr. Anthony Dadu from Department of Aquaculture, Ministry of Agriculture, Livestock and Fisheries, Tanzania. We would like to also thank Dr. Michael Phillips for his support to this research implementation process, and Dr Junning Cai and Jennifer Gee from FAO for sharing fish sector employment datasets and information.

## Author Contributions

**Conceptualization:** Chin Yee Chan, Nhuong Tran, Philippa J. Cohen.

**Data curation:** Chin Yee Chan, Nhuong Tran, Kai Ching Cheong.

**Formal analysis:** Chin Yee Chan, Nhuong Tran, Kai Ching Cheong.

**Funding acquisition:** Chin Yee Chan, Nhuong Tran.

**Investigation:** Chin Yee Chan, Nhuong Tran, Kai Ching Cheong.

**Methodology:** Chin Yee Chan, Nhuong Tran, Kai Ching Cheong, Timothy B. Sulser.

**Project administration:** Chin Yee Chan, Nhuong Tran, Kai Ching Cheong.

**Resources:** Chin Yee Chan, Nhuong Tran, Kai Ching Cheong.

**Software:** Chin Yee Chan, Nhuong Tran, Kai Ching Cheong.

**Supervision:** Chin Yee Chan, Nhuong Tran.

**Validation:** Chin Yee Chan, Nhuong Tran, Kai Ching Cheong, Timothy B. Sulser, Ahmed Mohamed Nasr-Allah.

**Visualization:** Chin Yee Chan, Nhuong Tran, Kai Ching Cheong.

**Writing – original draft:** Chin Yee Chan, Nhuong Tran.

**Writing – review & editing:** Nhuong Tran, Kai Ching Cheong, Timothy B. Sulser, Philippa J. Cohen, Keith Wiebe, Ahmed Mohamed Nasr-Allah.

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
