## [Decision Letter · Decision Letter 0]

15 Sep 2021

PONE-D-21-16152The future of fish in Africa: employment and investment opportunitiesPLOS ONE

Dear Dr. Tran,

Thank you for submitting your manuscript to PLOS ONE. After careful consideration, we feel that it has merit but does not fully meet PLOS ONE’s publication criteria as it currently stands. Therefore, we invite you to submit a revised version of the manuscript that addresses the points raised during the review process. The required revisions to the manuscript are between minor and major revision, but leans towards, minor revision. Looking forward to reading the revised version. 

We look forward to receiving your revised manuscript.

Kind regards,

Gideon Kruseman, Ph.D.

Academic Editor

PLOS ONE

Journal Requirements:

3. We note that Figures 1 and 2 in your submission contain map images which may be copyrighted. All PLOS content is published under the Creative Commons Attribution License (CC BY 4.0), which means that the manuscript, images, and Supporting Information files will be freely available online, and any third party is permitted to access, download, copy, distribute, and use these materials in any way, even commercially, with proper attribution. For these reasons, we cannot publish previously copyrighted maps or satellite images created using proprietary data, such as Google software (Google Maps, Street View, and Earth). For more information, see our copyright guidelines: http://journals.plos.org/plosone/s/licenses-and-copyright.

a. You may seek permission from the original copyright holder of Figures 1 and 2 to publish the content specifically under the CC BY 4.0 license.  

Reviewers' comments:

Reviewer's Responses to Questions

**Comments to the Author**

1. Is the manuscript technically sound, and do the data support the conclusions?

Reviewer #1: Yes

Reviewer #2: Partly

Reviewer #3: Yes

2. Has the statistical analysis been performed appropriately and rigorously? 

Reviewer #1: Yes

Reviewer #2: I Don't Know

Reviewer #3: N/A

3. Have the authors made all data underlying the findings in their manuscript fully available?

Reviewer #1: Yes

Reviewer #2: Yes

Reviewer #3: Yes

4. Is the manuscript presented in an intelligible fashion and written in standard English?

Reviewer #1: Yes

Reviewer #2: No

Reviewer #3: Yes

5. Review Comments to the Author

Reviewer #1: This is a well-written and balanced account to the much-neglected field of aquatic foods, fisheries and aquaculture systems in Africa. I really enjoyed reading it. Fact is that almost no sound data are available for capture fisheries and aquaculture investment costs , employment opportunities and farm-gate investment costs of key inputs. I hope more of these manuscripts are to follow.

I have made some additional grammatical and typographical corrections and restructured some sentences directly to the texts for consideration by the authors.

Finally, looking at the authorship and experience within it, I would like to see some clearer direction for decision makers on how the BAU and HIGH projection scenarios may be applied in management or policy decisions/implications. This would help a policy maker or even fisheries/aquaculture advocate be able to pull out a section and show how the data analyzed in this relevant is relevant to Continental National policy.

Reviewer #2: A good job. The attached review comment could contribute to its improvement especially the requested justifications. I wish to point that the article was of substantial information and should be published if adequate justifications are provided on the highlighted issues. The authors need to improve on tenses, grammar and the use of comma. There is the need to break quite a number of jaw breaking sentences.

Reviewer #3: “The future of fish in Africa: employment and investment opportunities” is an interesting and timely paper that addresses the future challenges and developments of the aquaculture and fisheries sectors in Africa. It does this using an existing tool, the International Model for Policy Analysis of Agricultural Commodities and Trade (IMPACT) developed by IPFRI (https://www.ifpri.org/project/ifpri-impact-model) to analyze two scenarios of developments in the fish production sector.

I suggest that the authors address a few issues with the paper ion its current form:

1. It is sometimes confusing whether the projections refer to the whole of Africa or the eight countries studied in detail. The current model is apparently an update from an earlier model based on detailed information from 8 African countries. The introduction then states that the projections are for eight African nations (line 100-101) while in several instances (Tables 3, 4 Line 241) the reference is made to the whole of Africa. This should be clarified

2. If projections are for the whole of Africa, the paper does not make clear how the detailed information is used to improved on the previous model. A flow chart detailing the steps made to do so is needed.

3. Some more detail on the main modelling assumptions is needed to be able to understand the scope and limitations of the analysis: the final; sentence in the discussion hints at these limitations – I suggest entering in more detail here and point out in what directions improvements are needed.

4. 76 experts were consulted in stakeholder sessions but its is not clear in what way this has improved the basic information collected for the model. For instance, what expertise was consulted? Which issues were discussed? How was consensus reached?

5. I have an issue with the conclusion that the results show that there is potential to sustainably increase capture fisheries. The high scenario, from which this conclusion derives I think, is based on a better accounting of current actual catches (rather than the estimated catches that are generally considered to be flawed) modelled as an increase in future catches. In other words, it basically assumes gradual better monitoring of otherwise stagnant (actual) fisheries production. In other words the model does not say anything about potential to sustainably increase catches. Many African fisheries experts consider todays freshwater catches as well as marine catches (especially in West Africa) unsustainable, though this can be disputed.

Detailed comments

Line 55-59 Rather awkward sentence. Please revise e.g. as

Whereas aquaculture is one of the fastest growing food production sectors globally, in Africa it supplies only 2.7% of the global share in 2019. Nevertheless, the African aquaculture sector is maintaining double digit average annual growth rates in the last two decades in response to continental market demand.

Line 112: “We apply the IMPACT model” - a few lines about the main features of this model are required for the reader to understand how projections are done.

Line 112: “This module”. Which module? “This” has no reference

Line 114: Previous application of the model: was this application limited to Africa or were these based on world historical trends?

Line 117: “The recent dataset” Which resent dataset is meant? Otherwise: “A recent data set compiled by ….”

Line 121 – Line 126: As the title of the paper is the future of fish in Africa, some discussion is required on how representative these 8 countries are for the rest of the continent, as the selection criteria as stated indicate a bias towards extremes in poverty, fish consumption and aquaculture growth

Line 124-126: Experts on what? Perhaps a list of (generalized) affiliations or expertise of the experts is useful?

Line 130 Table 1. Total fish production for Kenya, Tanzania and Uganda is 144+487+706 thousand ton = 1.337 million ton. The share of aquaculture production for these countries is 138.2 thousand ton leaving around 1.2 million ton of capture fisheries. Total catches of Lake Victoria alone over the three countries amounted to between 800 thousand to 1 million ton (between 2005 -2014, data LVFO), FAO Fishstat reports 1.1 million ton for total freshwater catches in 2019 for the three countries and 99.5 thousand ton Marine catches, which is around what is left after deducting freshwater catches and aquaculture. However, SeaAroundUs report reconstructed catches in 2018 as 140 thousand ton in Tanzania and Kenya 23 thousand ton totaling around 160 ton. Though this short review of data hat is available to me show that the estimates seem approximately what is agreed on it may be good to get a bit more insight in what basic data have been used. While for the HIGH scenario the Kolding et al.’s estimate of freshwater fisheries has been used, how is underreporting of marine capture estimates accounted for (for Tanzania and Kenya according to SeaAroudnUs captures in 2018 were 60% higher than the data used for BAU).

Line 142: “Environmental degradation continues” How is environmental degradation defined? What is meant by “at a slowing pace”? Compared to what?

Line 155: “Official statistics” as reported to FAO? Or in country official statistics? I.e. what data are used as data sets were also discussed with in country experts?

Line 151 - 154: “Exogenous productivity growth rates” What are these? “Exogenous” to what? It is not clear to me how this adjustment to reach an aquaculture output growth at 12.7%?

Line 161: “ Are accrued to the existing BAU projections” In other words the HIGH scenario assumes a gradual growth in freshwater towards the estimate by Kolding et al.? But this is an estimate of output the current freshwater capture fisheries. So in this scenario there is no room for additional growth in the freshwater sector. Also – African marine captures may also be underestimated (see SeaAroundUs estimates). How is that dealt with in this paper (for Ghana, Nigeria, Egypt, Tanzania, and Kenya).

Line 178-179 “…times employment multiplier.” Unclear – what multiplier is used here?

Line 236 – 241 – 277 In the two tables and text there is reference to African and Africa. How is the extrapolation from the 8 countries done to the African total? This is not clear from the methodology.

Line 357 – 359. “ These results show that there is potential to sustainably increase capture fisheries….” This is a strange conclusion as the HIGH scenario is based on a better accounting of current actual catches that – according to Kolding et al. paper cited, perhaps also SeaAroudnUs estimates – and is then not a gradual increase in actual catches from today’s underestimate. In other words the model does not say anything about potential to sustainably increase catches. On the contrary, though disputed, many African experts in fisheries consider todays freshwater catches as well as marine catches (West Africa) unsustainable.

Line 381 – 382 “ Global average labor productivity of 3.0 ton/worker” Apparently this is a weighted average over fisheries and aquaculture production. In an analysis of 16 African lakes Kolding, J., and van Zwieten, P.A.M. (2012) estimated a productivity of 3.0 ton per year per worker again indicating that the capture fisheries productivity of 1.7 over Africa may be way too low. (Kolding, J., and van Zwieten, P.A.M. (2012). Relative lake level fluctuations and their influence on productivity and resilience in tropical lakes and reservoirs. Fisheries Research 115-116, 99-109. doi:10.1016/j.fishres.2011.11.008)

Line 410 – 412. “We are unable to project investment costs needed to sustain capture fisheries, which mostly come from public funding and philanthropic investment” Not sure what investments are hinted at here? I guess this is investments in better monitoring and management capacity? What is meant by “philanthropic investment”: if by this is meant investment through e.g. Nature Conservation NGO’s then these are generally temporary investments that are not conducive to setting up permanent structures to maintain fisheries management, at most in capacity building.

Line 415-416. This sounds as a “must mention” line with limited or unclear links to preceding or following lines. The line also suggests that women are excluded from the employment opportunities mentioned in the previous sentence, which I don’t think will be or is the case. Please expand.

Line 423-426. As it is important to clarify the model assumptions in this study anyway, this final sentence should be expanded upon: what are the issues? What are avenues to improve the model based on your current insights.

6. PLOS authors have the option to publish the peer review history of their article (what does this mean?). If published, this will include your full peer review and any attached files.

Reviewer #1: **Yes: **Dr Kevin Obiero

Reviewer #2: No

Reviewer #3: **Yes: **Paul A.M. van Zwieten

---

## [Author Response · Author response to Decision Letter 0]

30 Oct 2021

RESPONSE TO REVIEWERS

We thank the reviewers’ comments and helpful advice. We responded to the critique, point-by-point as below, and made changes in the revised version (yellow highlighted).

Reviewer #1 Comments

This is a well-written and balanced account to the much-neglected field of aquatic foods, fisheries and aquaculture systems in Africa. I really enjoyed reading it. Fact is that almost no sound data are available for capture fisheries and aquaculture investment costs, employment opportunities and farm-gate investment costs of key inputs. I hope more of these manuscripts are to follow.

Q1: I have made some additional grammatical and typographical corrections and restructured some sentences directly to the texts for consideration by the authors.

A1: Thank you very much for your corrections. We have revised all your suggestions in the main text of the manuscript.

Q2: Finally, looking at the authorship and experience within it, I would like to see some clearer direction for decision makers on how the BAU and HIGH projection scenarios may be applied in management or policy decisions/implications. This would help a policy maker or even fisheries/aquaculture advocate be able to pull out a section and show how the data analyzed in this is relevant to Continental National policy.

A2: Thank you for your suggestions. The analysis of this work will be helpful for policymakers at the national and continental levels. For example, when the government prepares a Masterplan for the fish sector and estimates the future investment cost, it is vital to refer to the scientific references analyzed in this study to project the future prospects and challenges in production, consumption, trade, employment, etc. One published report [104] documented the uses of and outcomes from the Policies, Institutions and Markets (PIM)-supported foresight modeling research from 2012-2018. It covers usage by a wide range of stakeholders from across the Consultative Group on International Agricultural Research (CGIAR) system, other international organizations, academia, and national governments. 

This sentence is added in the Discussion section:

“As documented in a previous report [104], these results could be the practical usage by a wide range of stakeholders from international organizations, academic and national government.”

104. Lowder, S. K., Regmi, A. (2019) Independent Review: Assessment of outcomes based on the use of PIM-supported foresight modeling work, 2012-2018. Washington, DC: International Food Policy Research Institute (IFPRI). https://doi.org/10.2499/p15738coll2.133608

Reviewer #2 Comments

A good job. The attached review comment could contribute to its improvement especially the requested justifications. I wish to point that the article was of substantial information and should be published if adequate justifications are provided on the highlighted issues. The authors need to improve on tenses, grammar and the use of comma. There is the need to break quite a number of jaw breaking sentences.

Q1:The article is of importance to Africa’s future fish production. However, the following corrections/clarifications are needed. The objectives of the study is not clear under Abstract section. 

Line 27 had different font characteristic 

Line 30-31 not clear which one get 2.6 people, what system? And what specific contribution from each of feed, seed and labour 

A1: Thank you very much for your comments. The objectives of the study are added in the Abstract:

“To date, there are no estimates of investment and potential returns for domestic fish production in Africa. To contribute to policy debates about the future of fish in Africa, we applied the International Model for Policy Analysis of Agriculture Commodities and Trade (IMPACT) to explore two Pan-African scenarios for fish sector growth.”

The font characteristic in Line 27 is revised to the same “Arial” font size 11.

In Line 30-31, it means that for every person directly employed in the fish sector (both capture fisheries and aquaculture), 2.6 people will be indirectly employed. Drawing evidence from the existing literature, the study provides an aggregated estimation of employment effects. We are unable to project employment for each input sectors owing to missing information. We have revised the sentence as follow:

“Approximately 2.6 people will be employed indirectly along fisheries and aquaculture value chains for every person directly employed in the fish production stage.”

Q2: Line 51: In Africa? Line 55-59: has too long sentence and there are such all over the manuscript. These has to be corrected. The use of comma, tenses and grammar as observed for instance in lines 65, 93, 117, 178, 202 (growth?), 246, 266, 336 

A2: We have added “in Africa” to the sentence in Line 51: 

“Nevertheless, the current and future values of fish and aquatic foods in Africa are often overlooked in development research, policy and investment”.

The sentence from lines 55-59 was revised as follow for clarity:

“Whereas aquaculture is one of the fastest growing food production sectors globally [12, 13], Africa contributed only 2.7% to the global aquaculture share in 2019; though the growth rate is faster in the continent and maintained an average double digit annual growth rate over the past two decades”.

As your suggestions, we carefully checked and corrected grammar, tenses, and sentences in lines 65, 117, 178, 202, 246, 266 and 336.

Q3: There is need for ref in citing in some cases such as line 107. 

A3: A reference was added in line 107:

33. FAO. Low-Income Food-Deficit Countries (LIFDCs)-List updated June 2021. 2021. Available from: https://www.fao.org/countryprofiles/lifdc/en

Q4: Lines 100-107also 110-112 sounds line methods being presented in introduction

A4: Following your suggestion, we revised and moved the paragraph to the method section:

“We focused on eight African nations: Egypt, Ghana, Kenya, Malawi, Nigeria, Tanzania, Uganda and Zambia. These eight countries are home to 40% of Africa's total population but produce over 95% of aquaculture and 30% of capture fisheries production (by volume) in the continent in 2019. About half of fish consumed in Africa is by these eight countries, suggesting slightly higher per capita fish consumption rates than elsewhere in Africa [32]. Among these eight countries, Uganda, Tanzania, Malawi, Kenya, and Ghana are classified by Food and Agriculture Organization (FAO) as low-income food-deficit countries [33].”

Q5: It would be good to highlight the % of each country, the criteria for setting the ratio and how the experts look like (criteria for their selection /area of experts - aquaculture/fisheries? 

A5: We revised the sentences to highlight the proportion of countries and experts’ backgrounds:

“We consulted 76 experts from Egypt (43%), Nigeria (32%), Tanzania (15%), Zambia (4%), Ghana (2%), Kenya (2%) and South Africa (2%), during five stakeholder consultation workshops organized in Egypt, Nigeria, and Tanzania from 2017 to 2019. We sought the input of experts from government (27%), non-governmental organizations (47%), academia (13%), and private sector (13%) from different fields of expertise, covering fisheries, aquaculture, nutrition, gender, trade and economics to update and refine the model, explore alternative scenarios, validate projection results, verify the employment and investment dataset, and verify the post-model employment and investment estimation. The consensus had reached when no further comments from the stakeholders during the consultation process.”

Q6: Table 1 seems not clear. Justify why population growth ranged 5 years, GDP growth ranged 10 years, undernourishment only captured 2018, unemployment 2018. These variation need to be backed up with good reasons 

A6: We have revised table 1 for average annual growth ranging from the most recent ten years for population (2010-2020) and GDP (2010-2020). We also update the available latest year data for urban population (2020), GDP per capita (2020), undernourishment (2019), and unemployment (2020). 

Q7: Data being referred in Line 137-139 need to be cited 

A7: The two scenarios described in Line 137-139 are analysis of this study, therefore no citation in this paragraph. We have revised the sentence in Line 139:

“We had determined these trends from the regional experts we had gathered in the consultation workshops.”

Q8: Line141: what is the citation for the “Shared Socio Economic Pathways 2” 

Line 142- need to explain the assuming “slow pace environmental degradation. Explain what informed moderate challenges from climate change adaptation. 

A8: We have moved the citation next to Shared Socioeconomic Pathway (SSP) 2 and corrected the citation to the new reference of the original source for SSP. SSP 2 is a standard set of assumptions for socioeconomic development widely used in the modeling of future scenarios. All aspects of SSP 2 (environmental degradation, climate challenges, etc) are explained in O’Neli et al. 2017.

“In our BAU scenario, we use the Shared Socioeconomic Pathway (SSP) 2 [36], which assumes economic development continues but is not uniform, environmental degradation continues, but at a slowing pace compared to historical trends, and climate change presents moderate challenges to both adaptation and mitigation.”

36. O’Neill BC, Kriegler E, Ebi KL, Kemp-Benedict E, Riahi K, Rothman DS, van Ruijven BJ, van Vuuren DP, Birkmann J, Kok K, et al. The roads ahead: Narratives for shared socioeconomic pathways describing world futures in the 21st century. Global Environmental Change 2017;42:169-180. https://doi.org/10.1016/j.gloenvcha.2015.01.004

Q9: Line 147 seems to discuss aquaculture and fisheries together. Does it mean that data on the two were not separated. Check FAO data for separate figure for each of them. 

A9: The HIGH scenario we analyzed combines aquaculture and capture fisheries impact by using separate assumptions and data on the growth of aquaculture and compensation of under-reported capture fisheries production.

Q10: Line 153-154. You need to support the choice of the listed species with good reason. You would need to provide information on the % Africas aquaculture being contributed by mullet and pangasius? What brought their importance for projection in Africa. There are similar issues in line 163-164, 193 (they all need justification). 

A10: Thank you for your comments. In the IMPACT model, “Pangasius and other catfish” was defined as one fish group. To date, Africa only farms African catfish and does not farm Pangasius catfish. For clarity, we have revised the text and dropped Pangasius. Mullet is included in the projection as it is presently farmed in Egypt. We also describe the contribution in the percentage of farmed tilapia, catfish and mullet in Africa:

“This is achieved by adjusting the model’s exogenous productivity growth rates of the top five aquaculture producing countries in Africa (Egypt, Nigeria, Uganda, Ghana, and Zambia) for key selected species farmed in Africa (59% tilapia, 11% catfish and 11% mullet) from 2015-2050”

In Line 163-164, we have clarified the sentence:

“Under the HIGH scenario, we set a moderate optimistic annual income growth rate of 4.8% per year compared to SSP 2 of 2.9% under BAU.”

In Line 193, we have revised the sentence for clarity:

“For investment cost extrapolation, due to data limitations in capture fisheries, we focused explicitly on aquaculture alone to determine the size of investment needed to meet the BAU and HIGH projections of production.”

Q11: Line 200 rather use “broodstock” instead of “fertilized eggs”

A11: We have replaced “fertilized eggs” with “broodstock” as your suggestion.

Q12: Table 2; each column has cited many references but singler value was presented for each column, no variation/ standard deviation? In am curious why current values were not mentioned at all even in the discussion? Does it mean that the dynamics of these variants were held constant till now? 

A12: We estimated the baseline (2014 or 2016) of key aquaculture input parameters by referring to various references in eight African countries. We need a single value to project the future cost of key inputs by 2030 and 2050. We have presented this variant information from various references and validated the final consensus of a single value as listed in Table 2 with stakeholders during the consultation process. We extrapolate the future costs by using a single value in the base year that was converted to constant US$ in 2010 using the consumer price index.

We added one sentence for clarity:

“We present the variation of these inputs information in single value in Table 2 after validation via the stakeholder consultation process.” 

Q13: The study has too many limitations listed. It could be referred “preliminary”

A13: We agree with your comments. We have revised the last sentence in the conclusions section:

“This is a key and first preliminary analysis to look at macro-level employment and investment scenarios in fish sectors in Africa.”

Reviewer #3 Comments

The future of fish in Africa: employment and investment opportunities” is an interesting and timely paper that addresses the future challenges and developments of the aquaculture and fisheries sectors in Africa. It does this using an existing tool, the International Model for Policy Analysis of Agricultural Commodities and Trade (IMPACT) developed by IPFRI (https://www.ifpri.org/project/ifpri-impact-model) to analyze two scenarios of developments in the fish production sector.

I suggest that the authors address a few issues with the paper in its current form:

Q1: It is sometimes confusing whether the projections refer to the whole of Africa or the eight countries studied in detail. The current model is apparently an update from an earlier model based on detailed information from 8 African countries. The introduction then states that the projections are for eight African nations (line 100-101) while in several instances (Tables 3, 4 Line 241) the reference is made to the whole of Africa. This should be clarified.

A1: Thank you for your comments. The IMPACT fish model can project at global, continental, regional, and national level. This study project both the whole Africa and also the eight studied countries in Africa.

Q2: If projections are for the whole of Africa, the paper does not make clear how the detailed information is used to improved on the previous model. A flow chart detailing the steps made to do so is needed.

A2: We have added one sentence and a flow chart (Fig 1) to illustrate the steps of improvement of IMPACT fish model used to project the future of Africa’s fish sector on this study:

“The progressive improvement of IMPACT fish model used to project future Africa’s fish sector is illustrated” in Fig 1.”

Figure 1. Chronological model improvement and analysis using IMPACT fish model

Q3: Some more detail on the main modelling assumptions is needed to be able to understand the scope and limitations of the analysis: the final; sentence in the discussion hints at these limitations – I suggest entering in more detail here and point out in what directions improvements are needed.

A3: Thank you for your comments. We have listed the IMPACT fish modeling assumptions and limitations in our previous publication [17] under the conclusion section, as shown in the paragraph below. This study less emphasize on the model improvement but more focus on the post-model analysis on future employment and investment costs in Africa’s fish sector. To avoid the manuscript being too lengthy, we did not repeat these points in this study. 

“Modeling work for generating an evidence base for decision-making is crucial. While the IMPACT fish model has been greatly improved so that it can generate reasonable fish sector projections at both global and regional levels, the current setup suffers some limitations in terms of data and model structure. Available data is inconsistent and the lack of trade data at the desirable species classification leads to the inability to analyze bilateral trade flows, which makes the analysis of specific trade policy difficult without complementary work. Global markets with homogenized commodities is a necessary simplifying assumption that might be more contentious in the seafood market than, for example, in the cereal, meal or oil markets, where products are commoditized to a large degree. On the other hand, the sheer number of different fish species being caught and farmed requires some form of simplifications to be modelled at all. Other potential issues to address in the modeling framework include dealing with climate change and environmental stresses more explicitly, updating the underlying database with the latest available data and embedding the ability to develop new and alternative fisheries where they might not have previously been considered. The modeling efforts expose important data gaps and identifies the areas where improved data collection is needed.”

However, we have added the following sentences to highlight the issues and ways for improving fish foresight modelling and projection in the last paragraph in Discussion section:

“The current IMPACT model adopts highly aggregated with 16 fish categories in the supply side and 9 fish categories on the demand side. The model is calibrated using data in 2000 as a base year. This is quite out-of-date given that fisheries and aquaculture are complex and very dynamic, experiencing rapid growth over the last two decades. The IMPACT fish model uses generic assumptions to obtain parameters for specifying the fish sector equations, whereas fish and aquatic food systems are highly heterogeneous and complex. There are numerous fish types, classification schemes, and production methods. It is necessary to conduct disaggregated modeling studies for specific fish types to capture the diversity of trends within specific sub-sectors. Follow-up foresight modeling analysis and projection could address this disaggregation and complexity issues.”

Q4: 76 experts were consulted in stakeholder sessions but its is not clear in what way this has improved the basic information collected for the model. For instance, what expertise was consulted? Which issues were discussed? How was consensus reached?

A4: We revised the sentences to highlight the proportion of countries, experts’ background, issues being discussed and how consensus was reached during the consultation process:

“We consulted 76 experts from Egypt (43%), Nigeria (32%), Tanzania (15%), Zambia (4%), Ghana (2%), Kenya (2%) and South Africa (2%), during five stakeholder consultation workshops organized in Egypt, Nigeria, and Tanzania from 2017 to 2019. We sought the input of experts from government (27%), non-governmental organizations (47%), academia (13%), and private sector (13%) from different fields of expertise, covering fisheries, aquaculture, nutrition, gender, trade and economics to update and refine the model, explore alternative scenarios, validate projection results, verify the employment and investment dataset, and verify the post-model employment and investment estimation. The consensus had reached when no further comments from the stakeholders during the consultation process.”

Q5: I have an issue with the conclusion that the results show that there is potential to sustainably increase capture fisheries. The high scenario, from which this conclusion derives I think, is based on a better accounting of current actual catches (rather than the estimated catches that are generally considered to be flawed) modelled as an increase in future catches. In other words, it basically assumes gradual better monitoring of otherwise stagnant (actual) fisheries production. In other words the model does not say anything about potential to sustainably increase catches. Many African fisheries experts consider todays freshwater catches as well as marine catches (especially in West Africa) unsustainable, though this can be disputed.

A5: Thank you for your comment. We have revised the results conclusion to delete “sustainably increase” to “to sustain capture fisheries” :

“These results show that there is potential to sustain capture fisheries and expand aquaculture to meet the growing demand for fish in Africa.”

Q6: Line 55-59 Rather awkward sentence. Please revise e.g. as Whereas aquaculture is one of the fastest growing food production sectors globally, in Africa it supplies only 2.7% of the global share in 2019. Nevertheless, the African aquaculture sector is maintaining double digit average annual growth rates in the last two decades in response to continental market demand.

A6: Thank you for your correction. We have revised the sentence as you suggested.

Q7: Line 112: “We apply the IMPACT model” - a few lines about the main features of this model are required for the reader to understand how projections are done. 

Line 112: “This module”. Which module? “This” has no reference

Line 114: Previous application of the model: was this application limited to Africa or were these based on world historical trends? 

Line 117: “The recent dataset” Which resent dataset is meant? Otherwise: “A recent data set compiled by ….”

A7: Following your suggestions, we have revised the text describing the IMPACT fish model as follows:

In Line 112: “We apply the IMPACT fish model developed by International Food Policy Research Institute (IFPRI), which is a partial equilibrium economic model containing a system of equations for analyzing baseline and alternative scenarios for fish demand, supply, trade and prices at global, regional and country level in responding to future changes such as income, population and technological progress.”

In Line 114: “Previous application of the model by the World Bank in “Fish to 2030” report [29] used global historical data up through 2009 to develop business-as-usual (BAU) scenario.”

In Line 117: “To address these shortcomings, we re-calibrate the model with recent dataset and parameters of fish production, consumption, trade, population and GDP compiled from FAO, UN and IFPRI databases [4, 12, 30, 31].”

Q8: Line 121 – Line 126: As the title of the paper is the future of fish in Africa, some discussion is required on how representative these 8 countries are for the rest of the continent, as the selection criteria as stated indicate a bias towards extremes in poverty, fish consumption and aquaculture growth

A8: This study focus on the model projection and post-model analysis in Africa as a whole. Among 54 countries in Africa, we selected the top eight aquaculture and key capture fisheries producing countries to further analyze the future employment and investment costs. Among these eight countries, five countries (Uganda, Tanzania, Malawi, Kenya, and Ghana) are classified as low-income food-deficit countries. However, if we refer to the socio-economic indicators in Table 1, the performance of Egypt has a relatively above average both at global and at the continental level for extremes in poverty, fish consumption and aquaculture growth indicators. Therefore, the selection criteria still represent some country variations in poverty, per capita fish consumption, etc. Above all, our selection of these eight countries is not aim to represent the rest of the continent because we had conducted future projection and post-model analysis for Africa as a whole in this study.

Q9: Line 124-126: Experts on what? Perhaps a list of (generalized) affiliations or expertise of the experts is useful? 

A9: Thank you for your comment. We have provided experts affiliation and expertise as shown in A4 above.

“We consulted 76 experts from Egypt (43%), Nigeria (32%), Tanzania (15%), Zambia (4%), Ghana (2%), Kenya (2%) and South Africa (2%), during five stakeholder consultation workshops organized in Egypt, Nigeria, and Tanzania from 2017 to 2019. We sought the input of experts from government (27%), non-governmental organizations (47%), academia (13%), and private sector (13%) from different fields of expertise, covering fisheries, aquaculture, nutrition, gender, trade and economics to update and refine the model, explore alternative scenarios, validate projection results, verify the employment and investment dataset, and verify the post-model employment and investment estimation. The consensus had reached when no further comments from the stakeholders during the consultation process.”

Q10: Line 130 Table 1. Total fish production for Kenya, Tanzania and Uganda is 144+487+706 thousand ton = 1.337 million ton. The share of aquaculture production for these countries is 138.2 thousand ton leaving around 1.2 million ton of capture fisheries. Total catches of Lake Victoria alone over the three countries amounted to between 800 thousand to 1 million ton (between 2005 -2014, data LVFO), FAO Fishstat reports 1.1 million ton for total freshwater catches in 2019 for the three countries and 99.5 thousand ton Marine catches, which is around what is left after deducting freshwater catches and aquaculture. However, SeaAroundUs report reconstructed catches in 2018 as 140 thousand ton in Tanzania and Kenya 23 thousand ton totaling around 160 ton. Though this short review of data hat is available to me show that the estimates seem approximately what is agreed on it may be good to get a bit more insight in what basic data have been used. While for the HIGH scenario the Kolding et al.’s estimate of freshwater fisheries has been used, how is underreporting of marine capture estimates accounted for (for Tanzania and Kenya according to SeaAroudnUs captures in 2018 were 60% higher than the data used for BAU).

A10: Thank you for your detailed calculation on fish production. Insights from hidden harvest/under-reporting in capture fisheries studies was used to develop HIGH assumptions, allowing capture fisheries to reach intermediate level accounting of under-report data. In IMPACT fish model, the model structure did not further disaggregate capture fisheries to marine capture and inland capture. Therefore, the capture fisheries production will be at aggregate level.

Q11: Line 142: “Environmental degradation continues” How is environmental degradation defined? What is meant by “at a slowing pace”? Compared to what?

A11: SSP 2 is a standard set of assumptions for socioeconomic development widely used in the modeling of future scenarios. All aspects of SSP 2 (environmental degradation, climate challenges, etc) are explained in O’Neli et al. 2017. We have corrected the citation to the new reference for the original source of SSP and revised the sentence for clarity:

“In our BAU scenario, we use the Shared Socioeconomic Pathway 2 [36], which assumes economic development continues but is not uniform, environmental degradation continues, but at a slowing pace compared to historical trends, and climate change presents moderate challenges to both adaptation and mitigation.”

36. O’Neill BC, Kriegler E, Ebi KL, Kemp-Benedict E, Riahi K, Rothman DS, van Ruijven BJ, van Vuuren DP, Birkmann J, Kok K, et al. The roads ahead: Narratives for shared socioeconomic pathways describing world futures in the 21st century. Global Environmental Change 2017;42:169-180. https://doi.org/10.1016/j.gloenvcha.2015.01.004

Q12: Line 155: “Official statistics” as reported to FAO? Or in country official statistics? I.e. what data are used as data sets were also discussed with in country experts?

A12: We revised the sentence to clarify official statistics reported by FAO:

“For capture fisheries, FAO statistics [30] reported that, in 2017, Africa produced 7.0 million tonnes of marine capture fisheries and 3.0 million tonnes of inland fisheries.”

Q13: Line 151 - 154: “Exogenous productivity growth rates” What are these? “Exogenous” to what? It is not clear to me how this adjustment to reach an aquaculture output growth at 12.7%?

A13: Thank you for your comments. In the IMPACT fish model, aquaculture production is modeled as:

Aquaculture productivity growth rates are decided outside the model system of equations and exogeneous to the model. In this study, we adjusted exogeneous growth rates so that the aquaculture growth rate achieved 12.7% over the 2015-2030 period.

Q14: Line 161: “ Are accrued to the existing BAU projections” In other words the HIGH scenario assumes a gradual growth in freshwater towards the estimate by Kolding et al.? But this is an estimate of output the current freshwater capture fisheries. So in this scenario there is no room for additional growth in the freshwater sector. Also – African marine captures may also be underestimated (see SeaAroundUs estimates). How is that dealt with in this paper (for Ghana, Nigeria, Egypt, Tanzania, and Kenya).

A14: In line with your Q5, in this study, we assume that capture fisheries will grow to the level that under-reported and presented by Kolding’s studies [37]. This allows the capture fisheries to grow from the current official statistics reported to FAO to the capture fisheries output level, incorporating the unreported. Given that this unreported data are one-point estimation, further studies are recommended to examine if after incorporating unreported quantity, capture fisheries can grow further. Similar to Q10, HIGH sceanrio is using assumptions to compensate unaccounted capture fisheries (both inland and marine).

Q15: Line 178-179 “…times employment multiplier.” Unclear – what multiplier is used here?

A15: We have revised the sentence for clarity:

“We also take indirect employment equals direct employment (full-time equivalent number of jobs) times employment multiplier presented in Table 4.”

Q16: Line 236 – 241 – 277 In the two tables and text there is reference to African and Africa. How is the extrapolation from the 8 countries done to the African total? This is not clear from the methodology.

A16: We first update the production quantity to 2015 for all 54 Africa countries and calibrate to 2050 as describe in our previous publication, Fish to 2050 in Africa [1]. In this study, we select 8 countries to further update more detailed production (by 16 fish groups) and consumption quantity (by 9 fish groups) and further calibrate to 2050. Therefore, we are not using 8 countries to extrapolate for all Africa countries as a whole. In fact we conduct this continental and national level updates and analysis in two stages.

Q17: Line 357 – 359. “ These results show that there is potential to sustainably increase capture fisheries….” This is a strange conclusion as the HIGH scenario is based on a better accounting of current actual catches that – according to Kolding et al. paper cited, perhaps also SeaAroudnUs estimates – and is then not a gradual increase in actual catches from today’s underestimate. In other words the model does not say anything about potential to sustainably increase catches. On the contrary, though disputed, many African experts in fisheries consider todays freshwater catches as well as marine catches (West Africa) unsustainable.

A17: In line with Q5, we have revised the results conclusion to delete “sustainably increase” to “to sustain capture fisheries” :

“These results show that there is potential to sustain capture fisheries and expand aquaculture to meet the growing demand for fish in Africa.”

Q18: Line 381 – 382 “ Global average labor productivity of 3.0 ton/worker” Apparently this is a weighted average over fisheries and aquaculture production. In an analysis of 16 African lakes Kolding, J., and van Zwieten, P.A.M. (2012) estimated a productivity of 3.0 ton per year per worker again indicating that the capture fisheries productivity of 1.7 over Africa may be way too low. (Kolding, J., and van Zwieten, P.A.M. (2012). Relative lake level fluctuations and their influence on productivity and resilience in tropical lakes and reservoirs. Fisheries Research 115-116, 99-109. doi:10.1016/j.fishres.2011.11.008)

A18: Thank you for your comments. In Line 381-382, yes, the global average labour productivity of 3.0 ton/worker combines capture fisheries and aquaculture. The SOFIA 2018 [38] reported that the labour productivity for capture fisheries in Africa ranged from 1.46 to 1.77 between 2000-2016. The labour productivity of aggregate capture fisheries 1.73 ton/worker in 2016 [38] was used as our base year to extrapolate to future employment in capture fisheries. 

We appreciate your information very much from Kolding et al. 2012 to estimate the productivity 3.0 ton/worker in 16 African lakes. We recognize this reference showed higher labour productivity for inland capture fisheries in Africa. As explained in Q10, IMPACT fish model did not disaggregate capture fisheries into inland and marine. Therefore, we refer to labour productivity of aggregate capture fisheries (inland and marine) from the the SOFIA report in this study.

Q19: Line 410 – 412. “We are unable to project investment costs needed to sustain capture fisheries, which mostly come from public funding and philanthropic investment” Not sure what investments are hinted at here? I guess this is investments in better monitoring and management capacity? What is meant by “philanthropic investment”: if by this is meant investment through e.g. Nature Conservation NGO’s then these are generally temporary investments that are not conducive to setting up permanent structures to maintain fisheries management, at most in capacity building.

A19: Thank you. We have revised the sentence as below following your comments:

“Second, we are unable to project investment costs needed for capture fisheries monitoring, management and capacity building, which mostly come from public funding and development and conservation funding.”

Q20: Line 415-416. This sounds as a “must mention” line with limited or unclear links to preceding or following lines. The line also suggests that women are excluded from the employment opportunities mentioned in the previous sentence, which I don’t think will be or is the case. Please expand.

A20: Thank you for your comments. We deleted the sentence below to have a better link of preceding sentence:

“Enabling women to fully engage in and benefit from small-scale fisheries and aquaculture can boost production, reduce poverty, and improve food and nutrition security in Africa.”

Q21: Line 423-426. As it is important to clarify the model assumptions in this study anyway, this final sentence should be expanded upon: what are the issues? What are avenues to improve the model based on your current insights.

A21: We have added the following paragraph to highlight the issues and ways for improving fish foresight modeling and projection:

“The current IMPACT model adopts highly aggregated with sixteen fish categories on the supply side and nine fish categories on the demand side. The model is calibrated using data in 2000 as a base year. This is quite out-of-date given that fisheries and aquaculture are complex and very dynamic, experiencing rapid growth over the last two decades. The IMPACT fish model uses generic assumptions to obtain parameters for specifying the fish sector equations, whereas fish and aquatic food systems are highly heterogeneous and complex. There are numerous fish types, classification schemes, and production methods. It is necessary to conduct disaggregated modeling studies for specific fish types to capture the diversity of trends within specific sub-sectors. Follow-up foresight modeling analysis and projection could address these disaggregation and complexity issues.”

---

## [Decision Letter · Decision Letter 1]

29 Nov 2021

PONE-D-21-16152R1The future of fish in Africa: employment and investment opportunitiesPLOS ONE

Dear Dr. Tran,

Thank you for submitting your manuscript to PLOS ONE. After careful consideration, we feel that it is almost ready for publication. There is a final minor issue on page 23 lines 441-443, see comment of reviewer 3. Therefore, we invite you to submit a revised version of the manuscript that addresses the issue.

We look forward to receiving your revised manuscript.

Kind regards,

Gideon Kruseman, Ph.D.

Academic Editor

PLOS ONE

Journal Requirements:

Reviewers' comments:

Reviewer's Responses to Questions

**Comments to the Author**

1. If the authors have adequately addressed your comments raised in a previous round of review and you feel that this manuscript is now acceptable for publication, you may indicate that here to bypass the “Comments to the Author” section, enter your conflict of interest statement in the “Confidential to Editor” section, and submit your "Accept" recommendation.

Reviewer #1: All comments have been addressed

Reviewer #3: All comments have been addressed

2. Is the manuscript technically sound, and do the data support the conclusions?

Reviewer #1: Yes

Reviewer #3: Yes

3. Has the statistical analysis been performed appropriately and rigorously? 

Reviewer #1: Yes

Reviewer #3: N/A

4. Have the authors made all data underlying the findings in their manuscript fully available?

Reviewer #1: Yes

Reviewer #3: Yes

5. Is the manuscript presented in an intelligible fashion and written in standard English?

Reviewer #1: Yes

Reviewer #3: Yes

6. Review Comments to the Author

Reviewer #1: All previous comments have been addressed by the authors. The manuscript is now ready for publication

Reviewer #3: The reviewer comments were adequately addressed. One small suggestion for an edit: the final paragraph contains the following confusing newly added sentence:

"The current IMPACT model adopts highly aggregated with sixteen fish categories on the supply side and nine fish categories on the demand side." Perhaps "Given the high diversity in fished and cultured species in Africa, the current IMPACT model is highly aggregated with sixteen fish categories on the supply side and nine fish categories on the demand side." At least, if this was what was meant.

7. PLOS authors have the option to publish the peer review history of their article (what does this mean?). If published, this will include your full peer review and any attached files.

Reviewer #1: **Yes: **Dr Kevin Obiero

Reviewer #3: **Yes: **P.A.M. van Zwieten

---

## [Author Response · Author response to Decision Letter 1]

29 Nov 2021

A1: Thank you for your comments. We have revised the sentence as follow:

Given the high diversity in wild-caught and cultured fish species in Africa and worldwide, the current IMPACT model is highly aggregated with sixteen fish categories on the supply side and nine categories on the demand side.

---

## [Editor Report · Decision Letter 2]

7 Dec 2021

The future of fish in Africa: employment and investment opportunities

PONE-D-21-16152R2

Dear Dr. Tran,

We’re pleased to inform you that your manuscript has been judged scientifically suitable for publication and will be formally accepted for publication once it meets all outstanding technical requirements.

Kind regards,

Gideon Kruseman, Ph.D.

Academic Editor

PLOS ONE
---

## [Editor Report · Acceptance letter]

13 Dec 2021

PONE-D-21-16152R2 

The future of fish in Africa: employment and investment opportunities 

Dear Dr. Tran:

I'm pleased to inform you that your manuscript has been deemed suitable for publication in PLOS ONE. Congratulations! Your manuscript is now with our production department. 

Kind regards, 

on behalf of

Dr. Gideon Kruseman 

Academic Editor

PLOS ONE